# Projected loss of brown macroalgae and seagrasses with global environmental change

Federica Manca [1] ✉, Lisandro Benedetti-Cecchi [2], Corey J. A. Bradshaw [3,4], Mar Cabeza [1,5], Camilla Gustafsson [6], Alf M. Norkko [6], Tomas V. Roslin [7,8], David N. Thomas [1], Lydia White [6] & Giovanni Strona [9] ✉

Although many studies predict extensive future biodiversity loss and redistribution in the terrestrial realm, future changes in marine biodiversity remain relatively unexplored. In this work, we model global shifts in one of the most important marine functional groups—ecosystem-structuring macrophytes—and predict substantial end-of-century change. By modelling the future distribution of 207 brown macroalgae and seagrass species at high temporal and spatial resolution under different climate-change projections, we estimate that by 2100, local macrophyte diversity will decline by 3–4% on average, with 17 to 22% of localities losing at least 10% of their macrophyte species. The current range of macrophytes will be eroded by 5–6%, and highly suitable macrophyte habitat will be substantially reduced globally (78–96%). Global macrophyte habitat will shift among marine regions, with a high potential for expansion in polar regions.

Anthropogenic climate change is driving an unprecedented redistribution of the Earth's marine and terrestrial biota through extensive erosion, rearrangement, and shift of species ranges[1,2]. Such processes can alter existing biodiversity patterns and lead to the emergence of novel species assemblages, potentially disrupting important ecosystem services[2]. In the marine realm, brown macroalgae and seagrasses (hereafter 'macrophytes') play fundamental functional roles as both primary producers and habitat-forming organisms. Macrophyte-dominated habitats form extensive and highly productive biomes in shallow coastal areas worldwide, with brown macroalgal forests occupying an estimated 2.63 million km² (ref. 3), and seagrass meadows potentially covering up to 1.65 million km² (ref. 4). Canopy-forming brown macroalgae (including kelps and fucoids) and seagrasses enrich submerged vegetated habitats with complex, three-dimensional structures, offering shelter and food to many organisms[5], including threatened[6] and economically important species[7].

Given their essential role in marine environments, macrophytes sustain both small-scale and industrial fisheries, thereby underpinning a major supply of protein for millions of people and playing an important role in global food security, especially in low-income areas[7,8]. Seagrasses are of particular socio-economic and cultural importance for Indigenous people who depend on the resources and services provided by seagrass ecosystems[9,10]. Brown macroalgae are also harvested directly or farmed for food consumption, medical use, or to produce animal feed, fertilisers, biofuels, and other commercial products[11]. Besides ensuring coastal protection[12], brown algal canopies and seagrass stands also participate in marine biogeochemical cycles, with an estimated global primary production of 0.92 and 0.14–0.49 Pg

[1]Faculty of Biological and Environmental Sciences, University of Helsinki, PO Box 65 Viikinkaari 1, 00014 Helsinki, Finland. [2]Department of Biology, University of Pisa, CoNISMa, Via Derna 1, Pisa, Italy. [3]Global Ecology | Partuyarta Ngadluku Wardli Kuu, College of Science and Engineering, Flinders University, Adelaide, SA 5001, Australia. [4]Australian Research Council Centre of Excellence for Australian Biodiversity and Heritage (EpicAustralia.org.au), Wollongong, NSW, Australia. [5]Helsinki Institute of Sustainability Science, University of Helsinki, Helsinki, Finland. [6]Tvärminne Zoological Station, University of Helsinki, J.A. Palménin tie 260, 10900 Hanko, Finland. [7]Department of Ecology, Swedish University of Agricultural Sciences, Ulls väg 16, 756 51 Uppsala, Sweden. [8]Spatial Foodweb Ecology Group, Department of Agricultural Sciences, University of Helsinki, PO Box 27 Latokartanonkaari 5, 00014 Helsinki, Finland. [9]European Commission, Joint Research Centre, Ispra, Italy. ✉e-mail: federica.manca@helsinki.fi; giovanni.strona@ec.europa.eu

C year[-1], respectively[3,12]. Healthy brown macroalgal forests and seagrass meadows are important blue-carbon ecosystems that contribute to climate-change mitigation[12-14]. Together, the socio-economic services provided by brown macroalgae and seagrasses are estimated to be worth billions of dollars annually[15-18].

Multiple stressors from global change now threaten all these ecological and socio-economical services[19-21]. Dunic and colleagues[22] estimated that 19% of the surveyed seagrass area has already disappeared since 1880, and that the global loss of seagrass area for individual meadows is progressing at 1–2% year[-1]. Likewise, the geographical distribution of brown macroalgae has shifted[23-25]. The abundance of kelp (large brown macroalgal species of the order Laminariales) has slightly shrunk globally, but the magnitude and direction of change varies regionally, with declines in 38%, increases in 27%, and no evidence for change in 35% of the ecoregions where kelps are present[25]. Accelerating ocean warming has driven a poleward retreat of many species of brown macroalgae[23,26].

Although changes in macrophyte distribution might become even more prominent in the coming decades due to accelerating global environmental change, predictions of the future distributions of brown macroalgae and seagrasses are scant, in stark contrast to the many studies modelling the future distribution of terrestrial plants[27,28]. Currently available models projecting the future distribution of marine macrophytes apply to regional or local scales only, and/or to a limited set of species[29-33] (but see a recent global study[34]). Available studies focusing on the regional scale forecast substantial distributional shifts for both seagrasses[30,32] and brown macroalgae[31] by the end of this century. These shifts involve a high potential for local or regional extinctions[30] and poleward expansions[29]. However, there are no comprehensive, global-scale models of the future distribution of ecosystem-structuring marine macrophytes.

Brown macroalgae and seagrasses display distinct global distributions, both in terms of latitudinal patterns of biodiversity and spatial location of diversity hotspots[35-37]. Such differences arise both from long and complex biogeographic processes—such as distinct patterns of speciation and dispersal[38,39], and from taxon-specific ecological and environmental requirements. In general, temperature and light availability are commonly identified as important determinants of species distributions in both groups, followed by salinity, nutrient availability, wave energy, ice cover in polar regions, and the presence of suitable substrata[3,36,40,41]. Still, the relative importance and effects of these factors vary not only between the two groups but also within them. Thus, how the rapid shifts in these factors will reconfigure the distribution of habitat-forming marine macrophytes in future decades remains an open question, which calls for projections of both group-wide and species-specific responses to global change.

Here, we hypothesise that future increases in sea surface temperature will force brown macroalgae and seagrasses to retreat from lower to higher (and cooler) latitudes, albeit with substantial regional variation modulated by differences in salinity and surface primary productivity[42,43], light availability, water quality, and various other anthropogenic stressors acting at local and regional scales[25,44]. Climate-driven shifts in species ranges will likely change macrophyte community composition and local species diversity, with losses expected in tropical regions, where conditions are already close to the upper limits of thermal tolerance of many resident species[45]. By contrast, the projected reduction in sea-ice cover and increasing sea temperatures might promote an expansion of macrophyte distribution into polar regions[46], although potentially constrained by the availability of suitable substrata, declines in salinity, and increases in turbidity expected from sea-ice melting[47].

We test such hypotheses quantitatively by modelling yearly changes in the distribution of 207 seagrass and brown macroalgal species at the global scale from 2015 to 2100, thereby substantially expanding the taxonomic coverage and spatial extent of previous work. We show that by the end of the century, (i) the global species diversity of brown macroalgae and seagrasses will decline, with marked regional and latitudinal variation, (ii) coastal areas will become less suitable to ecosystem-structuring macrophytes, with global macrophyte habitat distribution shifting among marine regions, and (iii) both brown macroalgae and seagrasses will lose a large fraction of their present range, which will be barely compensated by expansions into new, suitable areas.

## Results

### Modelling individual macrophyte distribution

We modelled the global distribution of 207 marine macrophyte species (185 brown macroalgae and 22 seagrasses; Supplementary Fig. 1b, c; Supplementary Data 1–2) from 2015 to 2100 at a 0.5° × 0.5° latitude/longitude resolution and under three different greenhouse gas-emissions scenarios (Shared Socio-economic Pathways SSP2-4.5, SSP3-7.0, SSP5-8.5[48]). We first used a geometric criterion[49] to define the largest possible polygon where the species could occur (i.e., the *extent of occurrence*[50]), based on a large dataset of occurrence records[51]. We then applied a random forest algorithm[52] to obtain the *area of occupancy*[50] of each species by determining habitat suitability (ranging from 0 to 1) within the species' extent of occurrence, based on biologically relevant environmental and climatic variables (sea surface and air temperature, salinity, primary productivity, light, and sea-ice cover[3,36,40,41]; see Supplementary Fig. 1b and Methods for details). Our random forest models showed high accuracy, with an average out-of-bag validation score[52] of 0.987 ± 0.001 (± standard error; minimum = 0.773). On average, sea surface temperature, air temperature, and light had the highest influence[53] on predicting the distribution of both macrophyte groups (Supplementary Fig. 2).

### Present and future macrophyte species diversity

We stacked and summed individual area-of-occupancy maps[54] to obtain global maps of present and end-of-century macrophyte species diversity under different greenhouse gas-emissions scenarios (Fig. 1a, c; Supplementary Fig. 3a, c; Supplementary Fig. 4a, c; see Supplementary Fig. 5 for a map of linear species diversity). We obtained area-of-occupancy maps by determining habitat suitability within the boundaries of each species' present-day extent of occurrence (assuming no dispersal in areas where the species has never been recorded). Present hotspots of brown macroalgal diversity occur along the Pacific and Atlantic coasts of North America, the Atlantic coast of Europe, and the south-eastern coast of Australia (Fig. 1a). We identified hotspots of seagrass diversity in the north-eastern coast of Australia and the Caribbean Sea (Fig. 1c).

We explored end-of-century losses and gains of macrophyte species diversity by comparing maps of end-of-century and present diversity (Fig. 1b, d; Supplementary Fig. 3b, d; Supplementary Fig. 4b, d). Changes in marine macrophyte species diversity will follow different regional patterns. Under an intermediate carbon-emissions scenario (SSP3-7.0), our model predicts that the Pacific coast of South America will face the most severe losses in brown macroalgal diversity by 2100, with more moderate losses occurring in the eastern Indo-Pacific, in eastern Africa, and along the North Atlantic coast of North America, Europe, and Africa (Fig. 1b). Diversity loss in these regions appears particularly severe under the most pessimistic emissions scenario (SSP5-8.5; Supplementary Fig. 4b). Gains in brown macroalgal diversity are mainly clustered along the eastern coast of Africa and the Arabian Peninsula, the western coast of Africa, and northern Australia (Fig. 1b). Hotspots of seagrass diversity loss will occur mainly along the Pacific coast of North America, the Atlantic and Mediterranean coasts of Europe, Baltic Sea, Black Sea, Korean Peninsula, and the north-western and south-eastern coasts of Australia (Fig. 1d). Gains are mainly expected in the Caribbean Sea, and at intermediate to high latitudes in the Northern

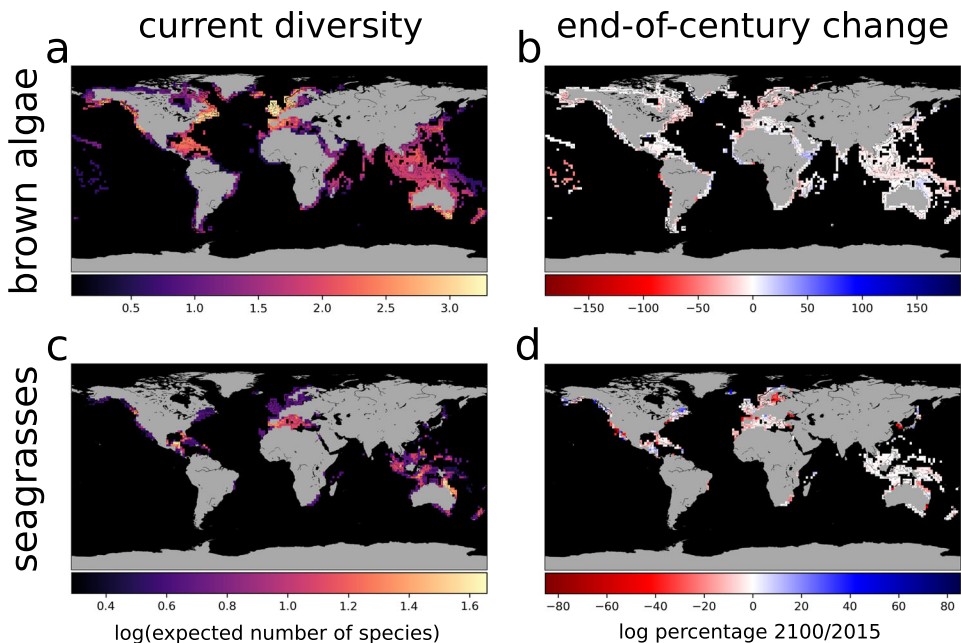

**Fig. 1 | Present distribution and projected end-of-century changes in global macrophyte species diversity. a**, **b** brown macroalgae, **c**, **d** seagrasses. We obtained present-day diversity (**a**, **c**) by stacking individual area-of-occupancy maps (see Methods). We computed end-of-century change (**b**, **d**) relative to 2015 using the $\log_e$ percentage change[84] ($100 \times \log_e[\mathrm{diversity}_{2100}/\mathrm{diversity}_{2015}]$) to show diversity losses and gains on a symmetrical scale. Gains in species diversity are shown in blue, and losses in red. The data to generate the maps include 185 species of brown macroalgae and 22 seagrass species. Maps refer to the intermediate emissions scenario (SSP3-7.0) and are upscaled to a 2° × 2° latitude/longitude resolution to ease visualisation. Analogous maps for more optimistic (SSP2-4.5) and more pessimistic (SSP5-8.5) scenarios are in the Supplementary Information (Supplementary Figs. 3 and 4). All maps were generated with the package *Basemap* in Python 3.

Hemisphere—particularly along the Pacific and Atlantic coasts of North America, and Iceland.

Of the localities where ecosystem-structuring macrophytes are present, globally 21.7% will lose ≥10% of their present brown macro-algal species diversity, and 17.4% will lose ≥10% of their current sea-grass species diversity (SSP3-7.0). Overall, losses of local species diversity are likely to outweigh gains, resulting in a net reduction of the average number of macrophyte species occurring at a given location by 2100 (Fig. 2a, b; Supplementary Fig. 6a, b; Supplementary Fig. 7a, b). Diversity loss is predicted to be more severe for brown macroalgae than seagrasses. Specifically, we expect to lose 4.4% ± 0.1% (± standard error) of local species of brown macroalgae by 2100 under an intermediate emissions scenario (SSP3-7.0), compared to a loss of 2.6% ± 0.3% of seagrass species (Fig. 2a, b). These figures reach 6.5% ± 0.2% and 6.7% ± 0.5%, respectively, under SSP5-8.5 (Supplementary Fig. 7a, b).

When considering latitudinal patterns of diversity change throughout this century (Fig. 2c, d; Supplementary Fig. 6c, d; Supplementary Fig. 7c, d), the loss of brown macroalgae diversity appears particularly stark beyond 40° N and in the entire Southern Hemisphere, while gains mainly occur between 0-20° N (scenario SSP3-7.0; Fig. 2c, d). The loss of seagrass diversity appears more severe between 25–40° in both hemispheres, while gains occur in the tropics (around 20° N and 20° S) and beyond 50° N in the Northern Hemisphere (scenario SSP3-7.0; Fig. 2c, d). Conversely, both groups are projected to face widespread diversity loss in the tropics under a more pessimistic emissions scenario (SSP5-8.5; Supplementary Fig. 7c, d).

**Future variation of global macrophyte habitat**

We used a random forest algorithm[52] to project present and future global habitat suitability for brown macroalgae and seagrasses at a resolution of 0.5° × 0.5° latitude/longitude for the period 2015–2100 and under different greenhouse gas-emissions scenarios[48]. Specifically, we modelled presence-absence of brown macroalgae (or seagrasses) per locality as a function of the same environmental and climatic variables we used to explore individual species' distributions (Supplementary Fig. 1a, see Methods). We considered macroalgae (or seagrasses) to be present in a given locality if represented by at least one occurrence (of any species in the dataset). Our purpose here was therefore to identify global-scale changes in the areas capable of hosting macrophytes, regardless of their identity or local diversity. This approach conveys substantially different information from that attained by summing species-specific suitability maps (Supplementary Fig. 8), not only from a conceptual perspective but also because global habitat suitability models identify an independent set of non-linear relationships linking environmental variables to macrophyte occurrence. Our global habitat suitability models had high accuracy, with an average out-of-bag validation score[52] of 0.817 ± 0.004 (±standard error; minimum = 0.806). The world's coastlines will become progressively less suitable for macrophytes, with highly suitable habitat (suitability $p > 0.9$; see Methods) expected to decline by 96.5% for seagrasses and 77.8% for brown macroalgae by the end of the century under an intermediate emissions scenario (SSP3-7.0; Fig. 3). Under the most pessimistic emissions scenario (SSP5-8.5, Fig. 3), the decline in highly suitable macrophyte habitat is projected to be more severe for brown macroalgae (81%), but not for seagrasses (92.1%).

Changes in macrophyte habitat suitability show different regional trends for the two groups (Fig. 4; Supplementary Figs. 9–11). Under the intermediate emissions scenario (SSP3-7.0), our model predicts a decline in the extent of suitable habitat ($p > 0.6$) for brown macroalgae in most marine regions (temperate North Atlantic, most of the Pacific and Indo-Pacific, temperate Australasia, temperate South Africa, tropical Atlantic, and temperate South America), with expansions limited to the Arctic, western Indo-Pacific and Southern Ocean (Fig. 4a). Conversely, the extent of seagrass-suitable habitat ($p > 0.6$) is expected to decline in the temperate North Atlantic, Indo-Pacific, temperate Australasia, temperate South-Africa and tropical Atlantic, with increases in the temperate North Pacific and Arctic (Fig. 4a). When considering highly suitable habitat ($p > 0.9$, Supplementary Fig. 9),

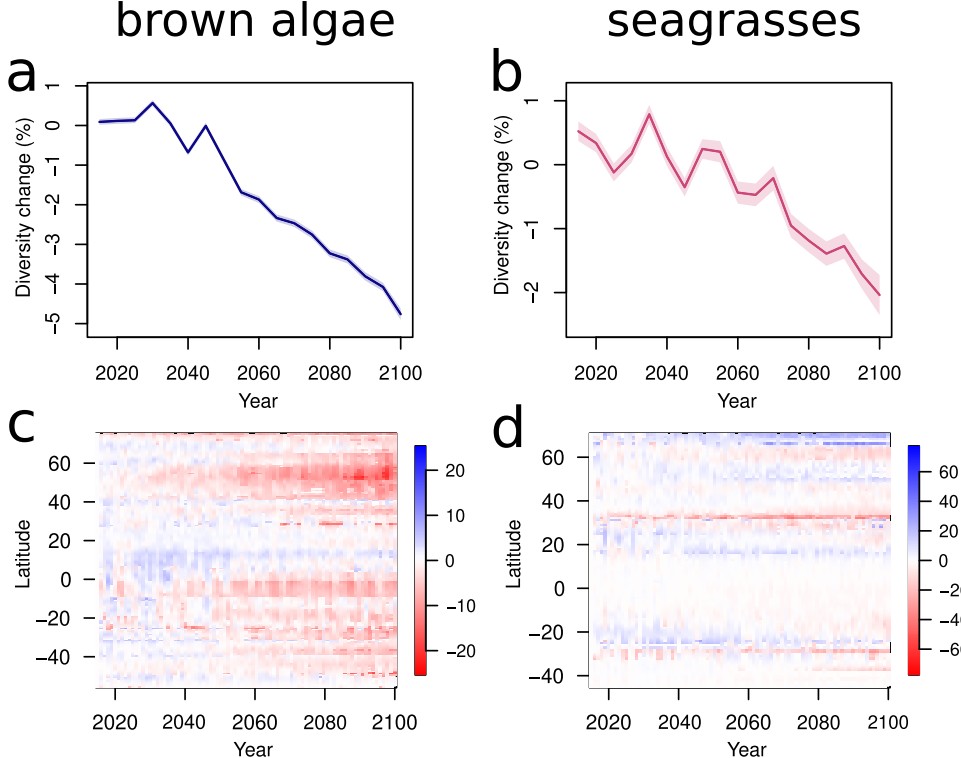

**Fig. 2 | Trajectories and latitudinal trends of future changes in macrophyte species diversity. a**, **c** brown macroalgae, **b**, **d** seagrasses. We computed change as $\log_e$ percentage change[84] in the 2100 diversity compared to 2015 diversity ($100 \times \log_e[\mathrm{diversity}_{2100}/\mathrm{diversity}_{2015}]$). The upper panels (**a**, **b**) report the mean trajectories (solid lines) and the 95% confidence interval (shaded area) in local macrophyte diversity (i.e., number of macrophyte species in every 0.5° × 0.5° latitude/longitude grid cell) relative to 2015. We aggregated data at 5-year intervals.

Lower panels (**c**, **d**) show expected future changes in diversity as $\log_e$ percentage change relative to 2015 diversity averaged across latitudes (0.5° × 0.5° latitude/longitude resolution). Gains in species diversity are shown in blue, and losses in red. Plots refer to an intermediate emissions scenario (SSP3-7.0), and analogous graphs for the more optimistic (SSP2-4.5) and more pessimistic (SSP5-8.5) scenarios are in the Supplementary Information (Supplementary Figs. 6 and 7).

declines are expected in all the marine regions where present, except for an expansion of brown macroalgae in the Arctic.

Regional variation in the extent of macrophyte habitat will translate into a substantial change in the share of suitable global macrophyte habitat ($p > 0.6$) in each marine region (Fig. 4b). Globally, the proportion of suitable habitat for brown macroalgae will mainly increase in the Arctic (from 2.7 to 8.4%), in the temperate North Pacific (from 19.7 to 26.5%), in the western Indo-Pacific (from 1 to 2.1%), while declining particularly in the temperate North Atlantic (from 41.2 to 37.9%), tropical Atlantic (from 6.3 to 0%), central Indo-Pacific (from 2.6% to 0.1%), and eastern Indo-Pacific (from 1.3% to 0.1%). Major expansions of the proportion of global suitable seagrass habitat will occur in the temperate North Pacific (from 10.5 to 24.7%) and Arctic (from 0 to 4.6%), while a marked decline is expected in the tropical Atlantic (from 13.2 to 1%) and central Indo-Pacific (from 11.9 to 4%). A decrease in the extent of macrophyte habitat in a region (Fig. 4a) does not necessarily correspond to a decrease in the relative proportion of globally suitable macrophyte habitat in the region (Fig. 4b) and vice versa, because the absolute global extent of macrophyte-suitable habitat is also expected to change by 2100 (Fig. 4b, square brackets).

### Erosion of average macrophyte area of occupancy
We explored future trajectories of average macrophyte area-of-occupancy extent by averaging the extension ($km^2$) of the individual area of occupancy of all brown macroalgae (185 species) and seagrasses (22 species) at yearly intervals from 2015 to 2100 and under different greenhouse gas-emissions scenarios (see Supplementary Fig. 1d and Methods for details). Note that changes in the extent of the

area of occupancy of each species can only be due to variation in habitat suitability within the boundaries of its present-day extent of occurrence (hence, we assume no dispersal in areas where the species has not yet been recorded). The analysis revealed similar outcomes for the two groups, with brown macroalgae expected to lose 5.8% ± 0.8% of their present area of occupancy by 2100, and 5.3% ± 2.9% for seagrasses under SSP3-7.0 (Fig. 5). We predict the loss to reach 11% ± 1.2% for brown macroalgae and 10.4% ± 4.1% for seagrasses under the most pessimistic scenario (SSP5-8.5; Fig. 5). Area-of-occupancy loss is unlikely to be distributed evenly among macrophyte species. Our models predict that most species will lose little of their present area of occupancy, while fewer species will experience severe losses by 2100 (Supplementary Fig. 12; Supplementary Table 1).

### Future opportunities
We explored future opportunities for expansion, assuming that all species could colonise novel suitable habitats beyond the boundaries of their current extent of occurrence to an extent proportional to the current ratio between occupied and suitable areas (see Methods). Under this assumption, only seagrasses might moderately expand their present average area of occupancy by 2100 (Fig. 5, dashed lines), with gains of 2.4% ± 2.6% (standard error) under an intermediate emissions scenario (SSP3-7.0) and 2.3% ± 3.2% under the more pessimistic SSP5-8.5. In contrast, no gains are expected for brown macroalgae under any emissions scenario (Fig. 5, dashed lines).

### Discussion
Our models show that climate change will have substantial, net detrimental impacts on the diversity and distribution of ecosystem-

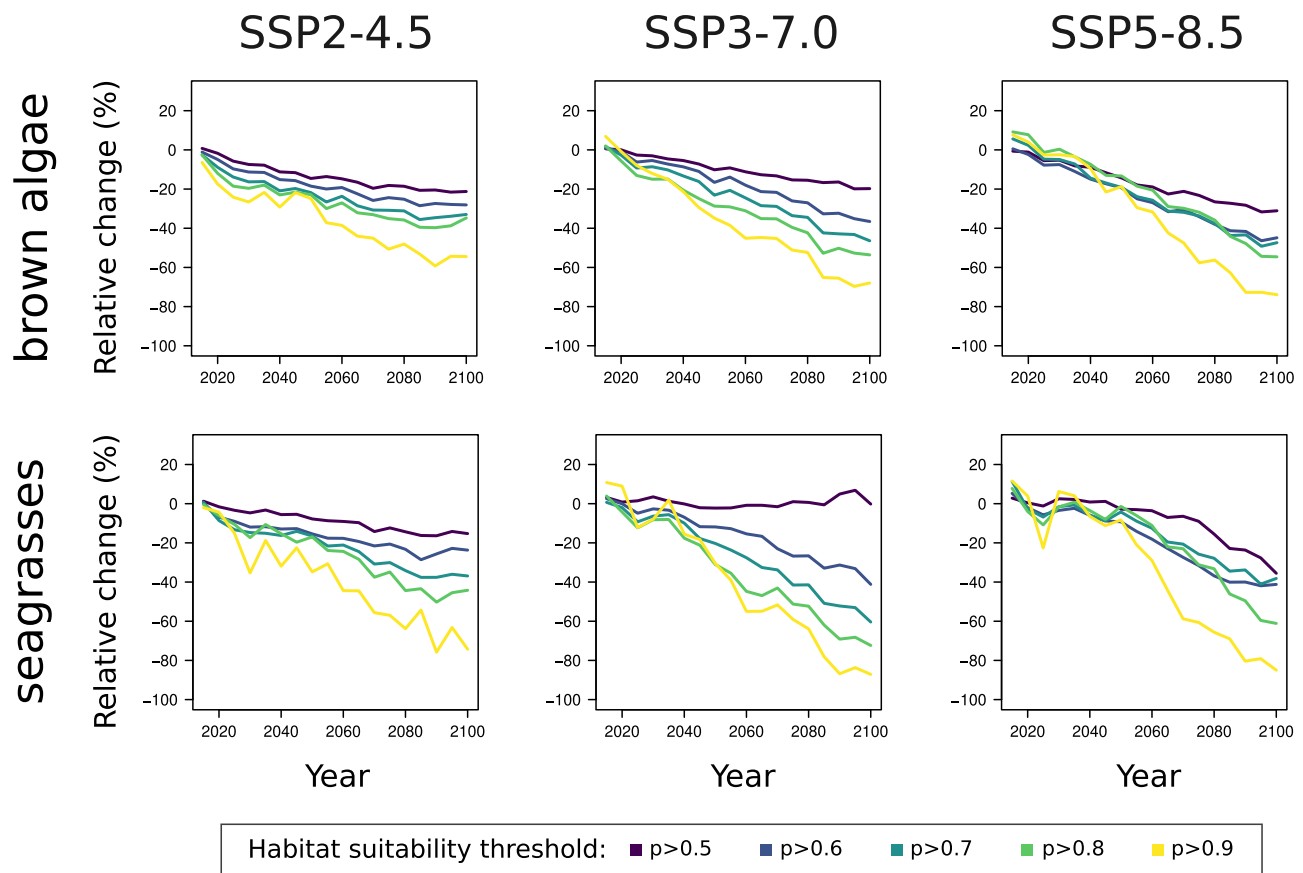

**Fig. 3 | Percent variation of the extent of suitable macrophyte habitat globally.** Brown macroalgae and seagrasses might lose >80% of their global highly suitable habitat ($p > 0.9$). We calculated global macrophyte habitat suitability using a machine-learning approach (see Methods) for the period 2015–2100. The trajectories show the percentage change in global habitat suitability relative to 2015 ($100 \times [area_{2100}-area_{2015}]/area_{2015}$), aggregated every 5 years. The colour scale indicates different probability ($p$) thresholds applied to the global suitability maps. Columns show different emissions scenarios and upper and lower panels show brown macroalgae and seagrasses, respectively.

structuring marine macrophytes. Projected responses of species to climatic and environmental drivers consistently point to a global decline in local species diversity, availability of suitable macrophyte habitat, and average macrophyte area-of-occupancy extent. Our results show contrasting patterns between the two macrophyte groups, with seagrasses being most affected by the loss of suitable habitat globally, and brown macroalgae experiencing higher losses of local species diversity.

In addition to modelling range contractions, we projected potential future opportunities for both macrophyte groups. Some species might in fact mitigate present area-of-occupancy erosion by colonising new areas that will eventually become climatically suitable[55]. However, the success of colonisation will depend on many factors, such as dispersal ability, water dynamics, ocean circulation[56], and ecological interactions[57]. Such complex interactions generate additional uncertainty regarding potential and realised future distributions. In our projections, we assumed that all species could exploit the opportunities provided by climate change proportionally to the current ratio between occupied and suitable areas. Such an assumption—a proxy for the various potential factors affecting the ability to colonise suitable areas—might be overly optimistic because it implies that all species will be able to occupy novel suitable areas instantaneously. This assumption necessarily oversimplifies the complex, long-term biogeographical mechanisms involved in these processes[38,39]. Within these caveats, our projections suggest that only seagrasses might partially offset area-of-occupancy erosion by expanding into new suitable areas.

Our projections of macrophyte diversity and habitat suitability vary substantially among ecoregions. In general, present patterns of brown macroalgal species diversity are consistent with those previously described by others[35,36]. Contrary to most marine coastal taxa[58], brown macroalgae show the highest diversity in temperate regions such as the north-western Pacific, the North Atlantic and south-eastern Australia. Conversely, the highest species diversity of seagrasses is mainly found in the tropics[37,59] (particularly in the Tropical Indo-Pacific bioregion). However, our map of current seagrass species diversity only partially overlaps with those of Short and colleagues[37,59] given that our models did not identify prominent diversity hotspots such as the insular Indo-Pacific and the coast of south-western Australia. The reason for this mismatch is that our analyses were focused on a specific set of seagrass species, for which sufficient occurrence data exist to make reliable predictions.

Climate change might substantially reconfigure global macrophyte diversity in the coming decades, with future changes exhibiting high regional and latitudinal variability. Latitudinal patterns of diversity change are not entirely consistent with our initial hypothesis of a prevalent loss of macrophyte biodiversity in tropical regions; while the diversity of brown macroalgae shows contrasting patterns in the tropics, seagrass diversity appears to remain mainly stable and increase in subtropical areas under an intermediate emissions scenario (SSP3-7.0). Widespread macrophyte diversity loss in the tropics only emerged under the more pessimistic scenario (SSP5-8.5). At intermediate to high latitudes—which host most brown macroalgal diversity at present—end-of-century climatic conditions might exceed the tolerance limits of resident species, thereby extensively reducing

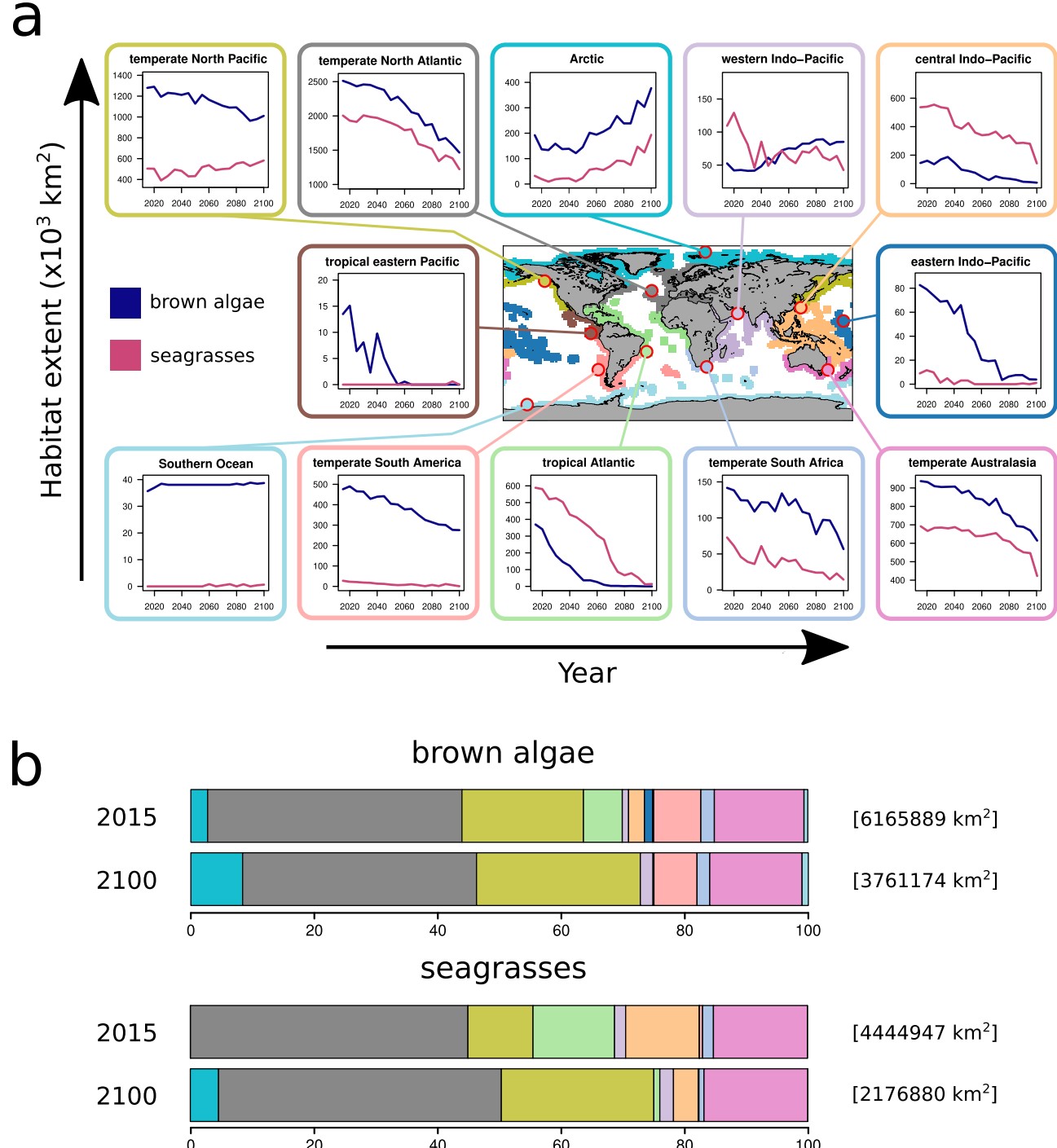

**Fig. 4 | Variation in the extent of suitable macrophyte habitat ($p > 0.6$) across marine regions for the emissions scenario SSP3-7.0.** We calculated global macrophyte habitat suitability using a machine-learning approach (see Methods) for the period 2015–2100, and applied a threshold $p = 0.6$ to ensure that all regions included at least one suitable cell. **a** Variation in macrophyte habitat extent (km²) for brown macroalgae (purple) and seagrasses (pink) within each marine region, aggregated every 5 years. **b** Comparison of the percentage of global suitable macrophyte habitat in each marine region between 2015 and 2100 for brown macroalgae (upper bar plots) and seagrasses (lower bar plots). Colours refer to marine regions as shown in (**a**). Square brackets show the total global suitable habitat extent. Analogous graphs for the more optimistic (SSP2-4.5) and more pessimistic (SSP5-8.5) scenarios are in the Supplementary Information (Supplementary Figs. 10 and 11). The map in (**a**) was generated using the package *Basemap* in Python 3.

brown macroalgal diversity. Seagrass diversity will also decrease in temperate regions, but increase at high latitudes. However, diversity losses are predicted with greater certainty than gains, because while the former can be determined by the loss of suitable abiotic conditions alone, the latter are also constrained by biotic and dispersal components that we did not incorporate in our models. Additionally, certain

species of brown macroalgae might respond to unsuitable abiotic conditions by migrating to greater depths rather than latitudinally[60,61]. These deep-water refugia could allow the persistence of brown algal forests in areas where area-of-occupancy contractions are projected. Vertical shifts in seagrass distribution are more unlikely, due to their generally higher minimum light requirements[40].

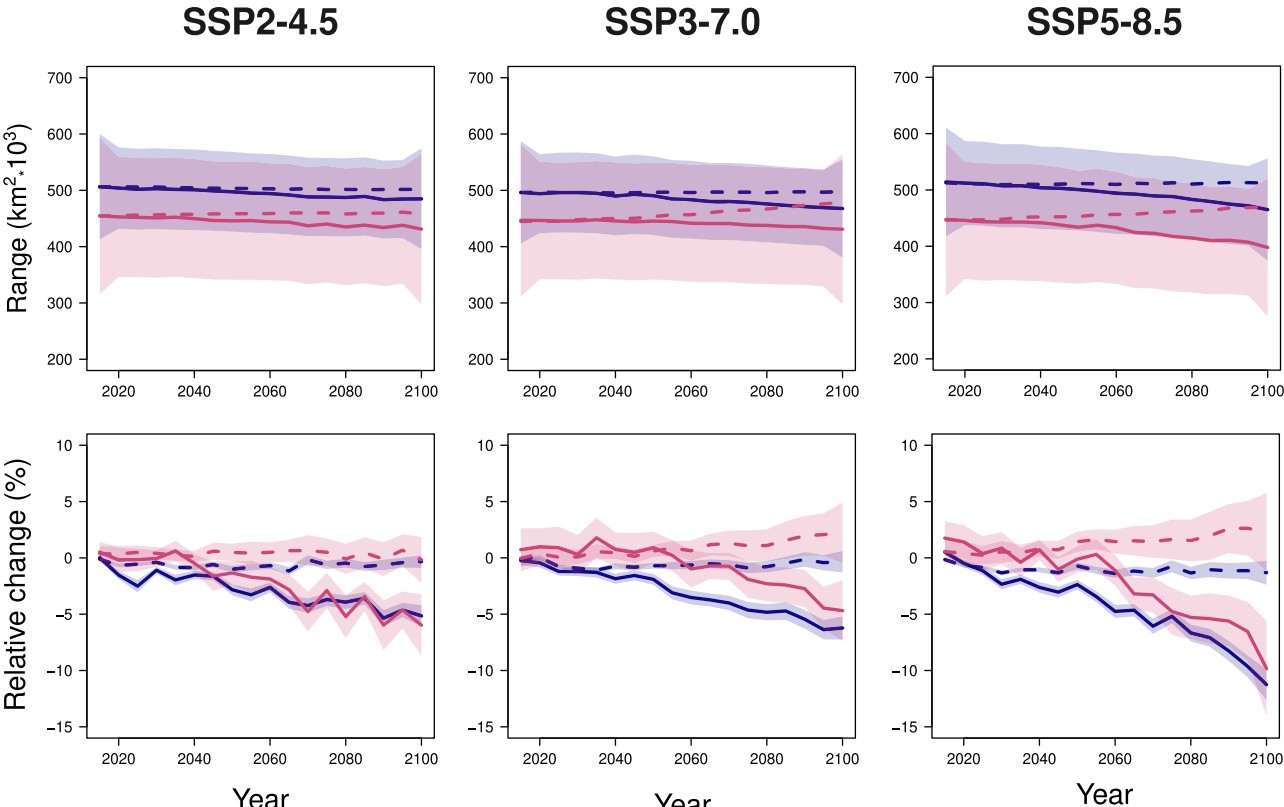

**Fig. 5 | Future variation in average macrophyte area of occupancy.** Brown macroalgae (purple; 185 species) are predicted to lose a higher percentage of their present area of occupancy than seagrasses (pink; 22 species) but expand less than seagrasses. Solid lines represent the mean variation in the macrophyte current area of occupancy (assuming no expansion beyond the species' extent of occurrence); dashed lines represent the mean expected change in the macrophyte area of occupancy assuming the current ratio between occupied and climatically suitable range will remain constant (*proportional expansion*; see Methods); this allows for expansion beyond the limits of a species' extent of occurrence. The shaded area represents a 95% confidence interval (not shown for proportional-expansion trajectories in the upper panel). The upper panel shows the expected change in km²; the lower panel shows the average percentage variation relative to 2015. Columns correspond to different emissions scenarios. We calculated habitat suitability for 2015–2100 using a machine-learning approach (see Methods), and area-of-occupancy trajectories by clipping individual habitat suitability maps with the species' present-day extent of occurrence obtained using a α-hull method (see Methods). We computed average trajectories over all macrophyte species and aggregated them every 5 years.

Several studies have documented poleward shifts in the distributional edges of marine taxa[1], including marine macrophytes[23,26], in response to warming, which are projected to continue over the coming decades[29]. We report a large expansion of suitable macrophyte habitat in the sub-Arctic and Arctic by 2100 in line with our hypothesis, observed trends[23], and future projections for the region[29]. We also projected a future increase in the share of global macrophyte habitat in the region. Conversely, our models did not predict a strong expansion of macrophyte habitat in the Southern Ocean, which is consistent with the independent expectation arising from the biogeographic isolation of the Antarctic continent[62], although dispersal to this region might occur via rafting for some species[63].

Despite high accuracy, our projections have some limitations. For instance, we only selected species with ≥10 occurrence records in the original dataset[51], a criterion that excluded 45% of the original number of species (47% brown macroalgae and 27% seagrasses). Our data and projections therefore focus on the more common and/or better-studied species, for which reliable predictions can be made. This choice explains discrepancies between the seagrass species diversity we predicted and those previously described[37,59]. In our data, relatively few seagrass species were included (30 seagrass species *versus* the approximately 72 seagrass species currently described—of which we modelled only 22). This causes a difference in the absolute species numbers represented in our predictions but also implies that the relative shifts projected come with high precision and confidence.

Our selection of environmental and climatic variables is a compromise dictated by data availability, the biological and ecological relevance of the hypothesised drivers, and the temporal and spatial scale of the analysis. Under all emissions scenarios, sea surface temperature and air temperature—together with surface incoming short-wave radiation—appeared to be the major determinants of macrophyte distribution. These results agree with the general expectation that distributional shifts of marine taxa mainly track changes in ocean temperature[1]. Although the variables we selected are established determinants of marine macrophyte distribution[3,36,41], several other abiotic factors not available at the necessary scale, resolution, and coverage might limit macrophyte distribution at local or regional scales—these include substrate availability, water turbidity and sedimentation, pollution, and habitat degradation[25,44]. In addition, the persistence of brown macroalgae and seagrass habitats in coastal areas will likely be affected by sea-level rise[64]—a variable we did not include in our models.

Furthermore, phenotypic plasticity induced by genetic and epigenetic modifications might allow some macrophyte species to cope with changing environmental conditions, making their responses to climate change challenging to predict[20,65]. The differing physiological responses to growth limiting factors between life-history stages of brown macroalgae offer another potential complication to the prediction of future distributions of this group[66,67]. In addition, biotic interactions can modify brown macroalgal and seagrass assemblages through top-down mechanisms[68,69], and will therefore likely play a role

in modulating the future distribution of both groups. For example, increased grazing pressure caused by the expansion of tropical herbivorous fishes has depleted brown algal cover in many temperate regions[68]. Interspecific interactions between individual macrophyte taxa—such as competition or facilitation—might contribute additional complexity[70]. However, our analyses are explicitly focused on a large, global scale—and the influence of biotic interactions on species distributions can be expected to diminish at large geographical scales and coarse resolutions (Eltonian noise hypothesis[71]). Thus, the extent to which biotic interactions will ultimately modify the projected patterns remains to be established. In the current context, considering this additional complexity at the taxonomic and spatio-temporal scales and the extent of our analyses would pose unsurmountable technical and analytical challenges.

As biotic factors identify potential filters further limiting species occurrence within their environmental niche, our predictions should be considered conservative. Nevertheless, our results provide strong quantitative support for the prediction that climate change will drive large changes in the biodiversity and distribution of ecosystem-structuring marine macrophytes. Given the foundational role of brown macroalgae and seagrasses in coastal ecosystems, the projected shifts will *ipso facto* modify coastal diversity and ecosystem functioning[72], and affect the essential ecosystem services these taxa provide, such as reducing the potential for carbon sequestration[73]. Considering the socio-economic value of seagrass and brown algal ecosystems, the projected changes might also erode human well-being and food security, especially in dependent coastal populations[7,10]. Overall, the substantial and geographically diverse redistribution of habitat-forming marine macrophytes projected in this study provides compelling evidence for the pervasive and intricate impacts of climate change on marine life[1,74].

## Methods
### Overview
We used extensive species occurrence data[51] and a set of environmental and climatic layers at the global scale to map both 'generic' brown macroalgae and seagrass habitats (i.e., habitat suitable to host any brown macroalgal or seagrass species, regardless of species identity), and the distribution of 207 individual macrophyte species (Supplementary Fig. 1, Supplementary Data 1–2). We derived maps for both the present and the future (from 2015 to 2100, at a yearly temporal resolution) under three greenhouse gas-emissions scenarios (Shared Socioeconomic Pathways[48] SSP2-4.5, SSP3-7.0, and SSP5-8.5), and explored future trajectories of change in macrophyte habitat extent and species diversity, as well as in individual species' ranges (defined as "area of occupancy"). We did all analyses in Python 3 and R version 4.2.1. We did spatial data manipulation and analysis using the *Geospatial Data Abstraction Library (GDAL)*, *Rasterio*, *Scipy*, and *Shapely* in Python 3.

### Species occurrence data
We compiled occurrence records of canopy-forming brown macroalgae and seagrasses from a large published dataset[51]. The dataset includes 2,751,805 georeferenced observations obtained from online repositories, herbaria, peer-reviewed publications, and citizen-science programmes that have been taxonomically standardised, dereplicated, and checked for accuracy. We used a pruned version of the dataset, filtered further to exclude records erroneously georeferenced to occur on land, in areas with unsuitable light conditions for photosynthesis of marine forests, or outside the known distribution of each species. Our final dataset includes 800,119 records spanning 1665 to 2018 for 30 species of seagrasses (families Cymodoceaceae, Hydrocharitaceae, Posidoniaceae, and Zosteraceae) and 349 species of brown macroalgae (orders Fucales, Laminariales, and Tilopteridales). From the latter, we also excluded two species of floating *Sargassum* (*S. fluitans* and *S. pusillum*).

### Environmental and climatic predictors
We selected a set of environmental and climatic predictors considered relevant for the distribution of both brown macroalgae and seagrasses: (*i*) sea surface temperature (°C), (*ii*) surface air temperature (°C), (*iii*) surface salinity (practical salinity units), (*iv*) surface primary productivity (organic carbon concentration in seawater, in g m$^{-3}$ day$^{-1}$), (*v*) light, in terms of surface incoming shortwave radiation (flux in the 0.2–4 μm wavelength band reaching a horizontal unit Earth surface, in W m$^{-2}$), irradiance at bottom (photosynthetically active radiation reaching the sea bottom, in E m$^{-2}$ year$^{-1}$), (*vi*) sea-ice cover (% of grid cell area covered by ice), and (*vii*) depth (m).

We hypothesised that changes in the selected variables are determinants of the future distribution of brown macroalgae and seagrasses, and in particular that (*i*) the projected global increase in sea surface temperature[42]—and surface air temperature for intertidal macrophytes[36]—will exceed the physiological thresholds of many brown algal and seagrass species, thereby reducing their productivity, survival, growth, and reproduction—particularly in populations at the trailing edge of their range[21,75,76]; (*ii*) the projected regional anomalies in surface salinity[43] will have varying effects on macrophyte survival, growth, and reproduction due to high interspecific variation in salinity tolerance[76,77]; (*iii*) regional increases in surface primary productivity associated with increased nutrient loading will reduce water clarity, thereby compromising macrophyte primary productivity[78,79]; and (*iv*) the projected reduction in sea-ice cover will increase the availability of substrata and sunlight for photosynthesis, potentially driving expansions of both groups in polar regions[46].

We obtained mean monthly sea surface temperature, air temperature, surface salinity, surface primary productivity, and sea-ice cover at a resolution of 0.5° × 0.5° latitude/longitude for 2015–2100 and under three emissions scenarios (SSP2-4.5, SSP3-7.0, and SSP5-8.5) from the Coupled Model Intercomparison Project Phase 6[80] (CMIP6; sea surface temperature, surface salinity, surface primary productivity, and sea-ice cover from www.dkrz.de/WDCC/ui/cerasearch/cmip6?input=CMIP6.ScenarioMIP.DKRZ.MPI-ESM1-2-HR.ssp845; air temperature from wdc-climate.de/ui/cmip6?input=CMIP6.ScenarioMIP.CNRM-CERFACS.CNRM-CM6-1-HR.ssp370). For incoming shortwave radiation at the surface, we averaged the monthly layers from 2019 to 2022 obtained from the Eumestat Climate Modelling Satellite Application Facility (wui.cmsaf.eu) at a resolution of 0.25° × 0.25° latitude/longitude. No future predictions of surface incoming shortwave radiation are available, so we maintained the same values for future predictions. We obtained depth from the 15 arc-second GEBCO interval grid (gebco.net). We downloaded data on the irradiance at the sea bottom (E m$^{-2}$ year$^{-1}$) from Bio Oracle (bio-oracle.org) at a 5 × 5 arc-minute resolution, considering the maximum values of radiance at the shallowest depth per cell. Where needed, we re-interpolated the original data to a resolution of 0.5° × 0.5° latitude/longitude.

### Macrophyte extent of occurrence
For each macrophyte species, we extrapolated the largest possible polygon where the species could occur at present (2015–2020) from point occurrences using a α-hull method at a resolution of 0.05° × 0.05° latitude/longitude, following the procedure described in Strona and colleagues[49]. We only selected species with ≥10 occurrence records, reducing the initial number of species (379; 349 brown macroalgae and 30 seagrasses) to 207 (185 and 22, respectively). We then started from a small α (0.001) to obtain a hull including most of the occurrences and progressively increased α by increments of 0.005. At each step, we computed the ratio between the reduction in hull area relative to the previous hull and the relative decrease in the number of occurrences contained in the hull (with respect to the total number of occurrences available for the target species). We selected the final value of α when the ratio became <10. We then re-interpolated each polygon to a 0.5° × 0.5° resolution and clipped them with the light-at-

bottom layer to exclude cells with limiting light conditions for macrophyte photosynthesis ($< 50 \, \text{E} \, \text{m}^{-2} \, \text{year}^{-1}$, as in ref. 51), and thus generated the present-day *extent of occurrence* for each species[50].

## Macrophyte habitat suitability (ecological niche model)

We used a random forest classifier (with the package *Scikit-learn* in Python 3) to model both (*i*) global macrophyte habitat suitability and (*ii*) species-specific habitat suitability at a resolution of $0.5° \times 0.5°$ latitude/longitude (Supplementary Fig. 1). Random forest is a machine-learning method based on an ensemble of bootstrapped decision trees[52]. Due to the high accuracy of predictions and the ability to handle complex interactions and collinearity among predictors[81], random forests have been increasingly used to model species distributions, including marine taxa[82,83]. For both *i* and *ii*, we used occurrence data from Assis and colleagues[51] and modelled the same set of $n = 207$ species selected with the $\alpha$-hull procedure. We modelled global macrophyte habitat suitability separately for brown macroalgae and seagrasses, considering presence cells as those hosting ≥1 species of brown macroalgae and seagrasses, respectively. To model species-specific habitat suitability, we considered presence cells as those hosting at least one occurrence record. In both cases, we randomly sampled pseudo-absences from cells with no occurrence records, maintaining the sample size equal to the number of presence cells.

We calibrated the models using the monthly layers of the environmental predictors: sea surface temperature, air temperature, surface salinity, surface primary productivity, and sea-ice cover for 2015–2020 under the three emissions scenarios (SSP2-4.5, SSP3-7.0, and SSP5-8.5), and the present values of surface incoming shortwave radiation and depth. In the individual models, we applied a variable-selection procedure where we iteratively removed the 10 least-important variables (starting from the full model), recomputing for each model the accuracy as an out-of-bag validation score[52] and variable importance as the mean accumulation of impurity decrease. We then selected the most accurate model. The procedure substantially reduced the number of predictors in each model ($55.8\% \pm 28.5$, on average). This also resulted in the exclusion of the most correlated independent variables in 82% of the models. We generated (*i*) global maps of brown algae and seagrass habitat suitability, and (*ii*) individual suitability maps for each of the 207 macrophyte species for 2015–2100 (under the three emissions scenarios), with probability of occurrence *p* ranging from 0 to 1.

## Analysis

We first modelled the future distribution of each of the 207 macrophyte species assuming no expansion beyond the boundaries of their present extent of occurrence. For this, we clipped individual habitat-suitability maps with the current extent-of-occurrence maps to obtain the *area of occupancy*[50] of each species for 2015–2100 and under the three emissions scenarios (Supplementary Fig. 1b). To generate global maps of species diversity, we stacked individual area-of-occupancy maps (Supplementary Fig. 1c, e). We estimated species richness as the sum of the probabilities of occurrence *p* of all the macrophyte species present in each $0.5° \times 0.5°$ grid cell[54]. We mapped end-of-century variation in global brown macroalgae and seagrass diversity as the $\log_e$ percentage[84] change between 2100 and 2015 ($100 \times \log_e[\text{diversity}_{2100}/\text{diversity}_{2015}]$) to show diversity losses and gains on a symmetrical scale. We also explored global trajectories and latitudinal trends of future changes in macrophyte diversity. For the latter, we calculated the $\log_e$ percentage change in diversity relative to 2015 for the years 2016 to 2100 averaged across latitudes (at a resolution of $0.5° \times 0.5°$ latitude/longitude).

We used global macrophyte habitat-suitability maps, clipped with the light-at-bottom layer to exclude cells with limiting light conditions for macrophyte photosynthesis[51] ($< 50 \, \text{E} \, \text{m}^2 \, \text{year}^{-1}$), to compute future trajectories of suitable habitat extension for brown algae and seagrasses both at a global scale and within each of the 12 world marine ecoregions[85] occupied by the modelled species (worldwildlife.org/publications/marine-ecoregions-of-the-world-a-bioregionalization-of-coastal-and-shelf-areas). To calculate global habitat extension, we converted global macrophyte habitat suitability maps into binary matrices by applying different probability thresholds ($p = 0.6, 0.7, 0.8, 0.9$) and summed the area (km²) of the resulting cells. We calculated variation in habitat extension relative to 2015 using the ratio $100 \times (\text{area}_{2100} - \text{area}_{2015})/\text{area}_{2015}$.

To calculate future trajectories of the average macrophyte area of occupancy from 2015 to 2100, we first multiplied, for each cell in the area-of-occupancy map, the probability of occurrence *p* by the corresponding cell extension (km²). We then summed the results to obtain the total extension of each species' area of occupancy. We averaged the extension of the area of occupancy of all 185 brown macroalgae and 22 seagrass species, respectively. We also explored potential alternative future trajectories of macrophyte area of occupancy under a hypothesis of *proportional expansion*. Because we did not explicitly model macrophyte dispersal, we assumed that all macrophyte species would be able to colonise novel suitable areas outside their present extent of occurrence by the same proportion represented by the current ratio between occupied and suitable areas. For each species, we calculated the ratio $R_{2015}$ between occupied and suitable (*s*) area for 2015 ($R_{2015} = \text{area of occupancy}_{2015}/s_{2015}$) and computed the potential area of occupancy under the proportional-expansion hypothesis for each following year $Y$ by multiplying $R_{2015}$ by the future suitable area $s_Y$ ($s_Y R_{2015}$).

## Reporting summary

Further information on research design is available in the Nature Portfolio Reporting Summary linked to this article.

## Data availability

The data used in this study were retrieved from the following sources: (1) macrophyte occurrence records: https://doi.org/10.6084/m9.figshare.7854767.v1[51]; (2) environmental predictors (mean monthly sea surface temperature, surface air temperature, surface salinity, surface primary productivity, and sea-ice cover): Coupled Model Intercomparison Project Phase 6 (https://doi.org/10.22033/ESGF/CMIP6.2450[80] and https://doi.org/10.22033/ESGF/CMIP6.4198[86]); (3) monthly incoming shortwave radiation: Eumestat Climate Modelling (CM) Satellite Application Facility (https://wui.cmsaf.eu[87]); (4) irradiance at sea bottom: Bio Oracle (https://bio-oracle.org[88]); (5) depth: GEBCO (https://doi.org/10.5285/c6612cbe-50b3-0cff-e053-6c86abc09f8f[89]). All the data used and generated in this study have been deposited in *Zenodo* with the identifier 10.5281/zenodo.10371401[90] [https://doi.org/10.5281/zenodo.10371401].

## Code availability

The code used in this work is available at https://doi.org/10.5281/zenodo.10907664[91].

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

## Acknowledgements

F.M. was supported by the Onni Talas Foundation and the Doctoral Programme in Wildlife Biology (LUOVA), University of Helsinki. L.B.C. acknowledges the contribution by the European Union's Horizon 2023 Research and Innovation Programme under grant agreement No. 101060072 ACTNOW. A.N., L.W. and C.G. acknowledge support from the Centre for Coastal Ecosystem and Climate Change Research (www.coastclim.org, University of Helsinki and Stockholm University).

## Author contributions

Conceptualisation: F.M. and G.S.; data analysis and visualisation: F.M. and G.S.; M.C. significantly contributed to results interpretation; F.M. wrote the original draft, with contributions from G.S. and C.B.; L.B.C., C.B., M.C., C.G., A.M.N., T.V.R., D.N.T., L.W. revised the manuscript and contributed to the final version.

## Competing interests

The authors declare no competing interests.
