## [Peer Review File · Nature Communications]

Projected loss of brown macroalgae and seagrasses with global environmental changeREVIEWER COMMENTS

Reviewer #1 (Remarks to the Author):

OVERALL

This study deals with the projected loss of biodiversity and suitable habitat by global warming of the most important foundation macrophytes of coastal marine ecosystems. For doing so, the extent of occurrence, habitat suitability and area of occupancy are calculated for 108 brown macroalgae and 15 seagrasses in the current climate (2015-2020) and under future IPCC scenarios, and individual patterns stacked to estimate species richness at each cell in the grid. The α -Hull method and Random Forest were performed and combined. Major conclusions are the decline in species richness, driven by significant contractions in species distributions that are not compensated by potential species expansions to new areas.

I believe the study of both observed and predicted patterns and trends of biodiversity globally is an important topic in Biology, Global Change Science, and Biogeography, as we expect major changes. However, the current ecological knowledge on such global biogeographic patterns is not related to the results of this study, and thus we cannot conclude how it is improved or completed previous knowledge. On the contrary, the authors pose that data may be not sufficient large for comparing with the diversity patterns described before (sixth paragraph in the discussion), and thus they do not include a discussion from the biogeographic perspective. Alternatively, they develop nicely the implications in global change scenarios for these two marine ecosystems, but I regret that in doing so they narrow the discussion of the results for a marine audience of experimental ecologists. There are not general statements or ecological theory other than potential impacts of climate change, and particularly the introduction and discussion sections do not deal with general ideas, as I argue below. For this reason, and because the biological database and species distribution modeling can be improved, I do not recommend the publication of this manuscript in Nature Communications.

LIST OF COMMENTS

The manuscript is focus from the beginning on the macrophyte-dominated habitats without framing the research in a more general biogeographic context of interest for a wide audience. In particular, the authors center the introduction on the mentioned ecosystems from the first sentence, without making any reference or comparison with other systems, marine or terrestrial, in regards of the biodiversity loss also experienced in those. I suggest start de introduction on the topic of global patterns of and losses of biodiversity worldwide and introduce the reader on the current knowledge on this topic. For this the authors may improve biological data by include the extensive records observed for macrophytes before 2015 (see comments below).

In relation to the above, I suggest fuse and shorten the first and second paragraphs of the introduction. I find well justified in the first paragraph the ecological relevance at a global scale of the studied systems, but do not see justification of a full paragraph on the contribution of macrophytes to carbon sequestration, a topic that is no mentioned afterwards. I suggest shortening drastically this paragraph.

Besides, the manuscript does not achieve general conclusions transferable to other ecological systems nor frame their findings on a biogeographic context in the discussion section. Please compare your results with those from Keith et al. (<https://doi.org/10.1111/geb.12132>), Tittensor et al. (<https://doi.org/10.1038/nature09329>), or Short et al. (<https://doi.org/10.1016/j.jembe.2007.06.012>), among others. Given the extensive literature and discussion on biodiversity global patterns and trends (mostly terrestrial), I encourage the authors to include this and similar literature using a biogeographic approach. This may help in understanding how the predicted loss of these canopy-forming species would impact the global marine biodiversity.

The authors develop Species Distribution Modeling (SDMs) using Random Forest, a method linking presence records and pseudo-absences with the environmental layers of relevant environmental predictors. As mentioned in the introduction, this modeling is important for predicting habitat suitability. This modeling tool, largely use by biogeographers, has not been framed in this context and sound references of the theoretical base are not included. The term SDM, or equivalent, is not mentioned throughout the manuscript. Please give a

brief explanation of what are these models and the state of the art, as there are not routinely used in marine systems.

Regarding the timeframe used in this study, I do not see justification what records previous to 2015 have not been used, as most observation of macrophytes distribution comprised in GBIF, OBIS, and Assis` databases were taken before. The extensive literature on biogeography was reviewed by Lüning in its popular book of 1990, there is large information on the distribution of the studied species that has been omitted without justification why.

Besides, the data used in this study are affected by the heat wave of 2003 and later, as well as by the baseline warming observed before this date. Therefore, some historical populations may be missing because of the warming, and thus the realized niche captured in the SDMs may not represent the full conditions the species is able to cope with. Please justify the timeframe of this study and account that data are already affected by climate change.

As the authors are probably aware, there is uncertainty in the ways biodiversity may be predicted in the future. In particular stacking predicted maps of areas-of occupancy do not inform on potential biological interactions and dispersal restrictions impeding the species to survive. The debate on how to improve predictions of biodiversity is open, and thus I suggest the authors to mention this potential limitation and properly account for this topic in introduction, methods, and discussion. See for example Zhang et al. (<https://doi.org/10.1111/ddi.12970>).

In relation to the above, distributional contractions by climate change linked to stressful climatic and physical conditions, and thus predicted with relatively confidence. On the contrary, expansions to areas that are becoming climatically suitable are dependent on the ability to disperse and overcome the inhibiting biotic interactions. This matter is explained by biogeographers using the BAM diagram (https://www.youtube.com/playlist?list=PLJSfnPII0T5i_6Svb4L4VO-yAHwaQVFeU). Given that macroalgae are not good dispersers and seagrasses rely on clonal growth, I believe that macrophytes will find difficulties in expanding their limits. Post filtering using information on

dispersal may improve the projections by SDMs (Anderson, <https://doi:10.1111/nyas.12264>). Instead, the authors argue that the current prevalence (ratio of presences/absences) of the species may capture dispersal limitations. I do not see how prevalence can inform on the elapsed time need for the species to reach the new area, that may be longer than the date of the projection. Post filtering the SDMs output using the dispersal rates of some model-species may help in identifying if the potential dispersal of relevant taxa may fall within the timeframe of the projections. I see clear theoretical differences between these two topics (dispersal Vs. prevalence-dominance) that are treated as a single response in paragraph starting in line 193, and in discussion in line 217. I believe prevalence is not a good proxy of dispersal, but please refer to previous references if so.

SPECIFIC ISSUES

Line 99- the authors are seeking for the “largest possible range where the species could occur” for calculating the extent of occurrence. For this I believe that data before 2015, particularly the extensive ones from the 70s, 80s and 90s, shall be considered. I see no justification for not using all the biological information available and thus suggest extending the database.

The lack of references in the “Results” and “Methods” sections is notorious. Line 98- lack of references on future climate scenarios, 100- lack of references on “extent of occurrence” and “area of occupancy” calculations, 104- lack of references of the validation method, 106- lack of references of the estimation of variance importance, 116- please add references to give support of the approach for predicting biodiversity. This is not solved in the “Methods” section where there are just two references on the niche modeling (numbers 28 and 76): 370- lack of references to random forest and SDMs, 388- lack of references of the protocol used, among others.

Line 110- How these results fit with the previous knowledge on the biodiversity global patterns of macrophytes and other relevant marine taxa?

Line 120- The “gains” are probably better linked to the eltonian niche than to the one

captured by SDMs. As argued above, gains cannot be given the same weight than losses, as are predicted with larger uncertainty. Please introduced this complexity in the ms. Are we really expecting biodiversity gains in the Arctic and sub-Arctic (coast of Iceland)? (Line 124). Besides, the species itself is not predicted to expand but its suitable habitat. Authors are projecting the species realized niche to the future rather than its presence. Please account for these limitations.

Fig 1. and similar- With such a low number of species of seagrasses (I understand is a small taxon in numbers), I do not see the need of use the logarithmic transformation of species richness, but the richness itself, as done in Short et al.

(<https://doi.org/10.1016/j.jembe.2007.06.012>). Besides, results of this study contradict those from Short and other authors. For example, the diversity in the Mediterranean seems greater than that of the Tropical Indo-Pacific contradicting previous knowledge. Moreover, Fig. 1 show maxima logarithmic values in the Mediterranean, contradicting the text in line 114.

Line 140- please clarify and provide references supporting this approach of used pooled data en SDMs, were the raw data of all species pooled as the input of biological records in a Random Forest algorithm? or alternatively, did the authors combined the habitat suitability values of each species in a single final value?. I regret the exact methodology is not clear in this paragraph. In line 296 it said "combine", please be precise.

Line 259- authors point here the potential importance of considered other factors not included in this study. I do not understand why this is given more weight that to the discussion of the responses to the environmental predictors that were considered, an aspect that is not discussed in deep.

Line 338- How you avoid autocorrelation between environmental predictors? this is well known to be an potential issue in SDMs and other multivariate approaches (Elith, <https://doi.org/10.1111/j.2041-210X.2010.00036.x>). The monthly means of the environmental parameters used in this study show necessary high autocorrelation. Please account for this important issue when modeling.

Line 381- the models are calibrated for the recent-present day layers but cannot be trained using future conditions (SSP2-4,5, SSP3-7.0, AND SSP5-8.5) contrary to what is suggested in Methods. Training and predicting are mixed up and confused in the text: the models are first trained to the current-recent conditions and afterwards the final algorithm applied to future scenarios, please explain separately these two phases of the modeling process.

Please end discussion making generalizations that arise necessary from comparisons with previous work. With the exception of the topic of global decline of macrophytes due to climate change, the authors do not include other general conclusion on how these results contribute to our understanding of the patterns and trends of global diversity. I strongly believe the manuscript has to give same weight to climate change science and biogeography.

Reviewer #2 (Remarks to the Author):

The authors use georeferenced species occurrence data for brown algae and seagrasses derived from online repositories as well as environmental data that are known to limit the distribution of these species. They then modelled how these environmental factors will change under future climate scenarios and what this would mean for the global distribution of these two groups of macrophytes. I am not a modeller so I can't comment on the validity of methodological approaches used, but the environmental data used to drive their models are appropriate and the outcomes of the analyses seem logical and align with similar regional studies. Overall this paper provides the first global analysis of how these two important groups of macrophytes are likely to shift their ranges into the future. As with any modelling approach there are a number of caveats that mean the predictions made in this manuscript may not come to pass. On the whole these are the same for any similar study and there is little that the authors could do, but in places I think the caveats need to be more strongly acknowledged. The authors also highlight that other studies separate model intertidal and subtidal species differently because drivers of their distribution are different. This study does not separate intertidal and subtidal species, but no justification is given for why this is appropriate in this study. This is a concern and I would argue that the authors should model these species separately as other studies have for the reasons stated or

provide an evidence based justification for why it is okay in this instance. The number of species modelled (123) and the geographical coverage (12 eco-regions) make this study novel. I enjoyed reading the manuscript, but feel that authors need to explain how their approach of not separating intertidal and subtidal species is an appropriate approach given the different drivers of distribution for these species and therefore at this stage I believe that this manuscript can not be accepted for publication in its current state.

Minor comments

Lines 35 & 36 The references should come after the areal extent of the two habitats

Line 54 It should be made clear that both DOC and POC can be transported to deep marine sediments

Line 64 Krumhansl et al 2016 PNAS 113: 13785-137990 should be cited at the end of this line and it might also be worth stating how many ecoregions have reported losses

Lines 67-70 I suggest these sentences are restructured as the first suggests that there are no studies, when there are regional studies and then the second sentence acknowledges that this is the case.

Line 78 It seems strange that only salinity and primary productivity are noted here. Light will be a key driver of these species distributions as will other interacting anthropogenic stressors. This is acknowledged in the discussion, but should be incorporated here as well.

Lines 81-82 Some macrophytes are likely to expand into the Arctic with reductions in sea-ice cover, but the picture is more complicated as outline in Filbee-Dexter et al Global and planetary Change 172: 1-14 review paper. I suggest caveating this sentence to acknowledge the more complex picture.

Line 123 There is also a hotspot of loss on the west coast of Australia that deserves a mention, particularly as this area supports a high diversity of seagrass species.

Lines 129-131 I suggest stating what the losses under the other climate change scenarios are

and refer the reader to the SOM

Line 133 Add negatively before impacted

Line 134 I suggest adding that these changes are particularly stark from 20 degrees north

Line 135-137 I suggest it might be worth highlighting the gains and then losses around 20 degrees south

Lines 164-166 It isn't clear to whether the values in parentheses are the increases/ decreases between 2015-2100 or the range of values the models provided. Please clarify

Lines 241-243 From my understanding there is increasing concern of brown macroalgae reaching the Antarctic via the rafting of, particularly species with air bladders, that have been dislodged bearing reproductive material. It may be worth acknowledging this as a method of dispersal, but not one that would be picked up with this modelling approach.

Line 245 It is only under the worst case scenario that any brown algal species losses most of their range. This should be acknowledged.

Line 249 I believe that *Macrocystis pyrifera* has a listing designation in Australia that might be worth acknowledging if this species is included in the analysis

Figure 2 It should be stated in the figure legend what the blue and red represent in terms of gains and losses

Figure 4 If this was upped to a higher probability how many regions would be excluded. I would be tempted to use as high a probability as possible e.g. $p > 0.9$ as described in the text around Fig 3 and only use this lower value for the regions with limited suitable cells unless this is the case across all/majority of regions and if so this needs to be stated.

SOM Fig 3 and 4 are only discussed in the discussion and all SOM figures should be

numbered in the order in which they appear.

Reviewer #3 (Remarks to the Author):

This study uses habitat suitability modelling and climate projections to show how global distributions of large brown macroalgae (LBMs) and seagrasses are likely to change over the next 70 years or more. The approach is solid, if a little opaque in parts, with the findings only completely clear after a close study of the methods. Species with sufficient records to establish credible global distributions are modelled individually and the results combined to show how species diversity and total suitable habitats might change. The second part of the work models all seagrasses and LBMs together to produce similar projections to those from the combined individual species models, without adding much to the work. Dropping the total macrophyte modelling would allow for greater focus on how the findings are supported by the methodology. The changes in macrophyte results shown in Figs 3 to 5 could be produced from the summation of the species-specific models.

Some confusion for this reader was generated by the very similar-sounding terms for quite different measures of species ranges and suitable habitats. 'Extent of occurrence' covers the range of observations of each species, while 'area of occupancy' refers the overlap of 'extent of occurrence' and modelled maps of present-day and predicted future suitable habitat (lines 389-393). Future 'area of occupancy' can never therefore never exceed present-day 'extent of occurrence'. 'Area' is also used interchangeably to mean the summed extent of a species distribution and the distribution itself.

Models of future distributions assumed no expansion of ranges (line 388). Adopting this decision meant that the focus was only the loss of present-day ranges, with largely negative impacts on future diversity. For these largely temperate species with no tropical counterparts to replace them with warming, declining diversity would be expected in their present-day temperate diversity hotspots. But the diversity change calculated here (lines 395-396) does not take account of potential gains from range expansions of warm-affinity species within temperate zones, inflating the likely losses in diversity.

'Proportional expansion' (lines 411), the assumption that species will continue to occupy a constant proportion of the predicted globally suitable habitat, is a fair enough proposition and better than that newly suitable habitat will be instantly occupied once it becomes available through climate change.

More detailed comments follow:

lines 26-27. Losses. It is not yet clear at this point that gains in range and suitable habitats are excluded, and expansions focussed on macrophyte habitat in general.

lines 34-75. This is a nice summary of the literature on the main drivers of change in these groups, but the number of references could be reduced.

line 120. The odd gains of species diversity for LBMs in the northwest Indian Ocean are hard to understand (Fig. 2b). Are existing tropical species predicted to expand to areas previously unoccupied? Similarly, gains for seagrasses east Africa (Fig. 2d) need some explanation.

Figure 3. The greater losses of seagrass habitat extent under the less-extreme climate scenarios of SSP2 and SSP3 than SSP5 are difficult to reconcile. Why is that happening?

Figure 4. Information presented here is essentially similar to Figs 1 and 2, with declines in tropical regions, stasis in cold temperate regions and increases in polar regions where suitable habitats may exist – here the Arctic only.

Figure 5. As before, distinction between contraction of existing distributions (solid lines give changes in area of occupancy) and potential expansion into newly suitable areas (dashed lines) makes it difficult to see a net gain or loss in LDMs and seagrasses.

Reviewer 1

I believe the study of both observed and predicted patterns and trends of biodiversity globally is an important topic in Biology, Global Change Science, and Biogeography, as we expect major changes. However, the current ecological knowledge on such global biogeographic patterns is not related to the results of this study, and thus we cannot conclude how it is improved or completed previous knowledge. On the contrary, the authors pose that data may be not sufficient large for comparing with the diversity patterns described before (sixth paragraph in the discussion), and thus they do not include a discussion from the biogeographic perspective. Alternatively, they develop nicely the implications in global change scenarios for these two marine ecosystems, but I regret that in doing so they narrow the discussion of the results for a marine audience of experimental ecologists. There are not general statements or ecological theory other than potential impacts of climate change, and particularly the introduction and discussion sections do not deal with general ideas, as I argue below. For this reason, and because the biological database and species distribution modeling can be improved, I do not recommend the publication of this manuscript in Nature Communications.

RESPONSE #1: We have now framed our study in a broader biogeographic context both in the Introduction (L34–37):

“Anthropogenic climate change is driving an unprecedented redistribution of the Earth’s marine and terrestrial biota through extensive erosion, rearrangement, and shift of species ranges^{1,2}. Such processes can alter existing biodiversity patterns and lead to the emergence of novel species assemblages, potentially disrupting important ecosystem services².”

L79-89:

"Brown macroalgae and seagrasses display distinct global distributions, both in terms of diversity gradients and spatial location of diversity hotspots^{35–37}. Such differences arise both from long and complex biogeographic processes^{38,39}, and from taxon-specific ecological and environmental requirements. In general, temperature and light availability are commonly identified as important determinants of species distributions in both groups, followed by salinity, nutrient availability, wave energy, ice cover in polar regions, and the presence of suitable substrata^{3,36,40,41}. Still, the relative importance and effects of these factors varies not only between the two groups, but also within them. Thus, how the rapid shifts in these factors will reconfigure the distribution of habitat-forming marine macrophytes in future decades remains an open question, which calls for projections of both group-wide and species-specific responses to global change."

and the Discussion (L269-279):

“Our projections of macrophyte diversity and habitat suitability vary substantially over the global range. In general, present patterns of brown macroalgal species diversity are consistent with those previously described by others^{35,36}. Contrary to most marine coastal taxa⁵⁸, brown macroalgae show highest diversity in temperate

regions such as the north-western Pacific, the north Atlantic and south-eastern Australia. Conversely, the highest species diversity of seagrasses is mainly found in the tropics^{37,59} (particularly in the Tropical Indo-Pacific bioregion). However, our map of current seagrass species diversity only partially overlaps with those of Short and colleagues^{37,59} given that our models did not identify prominent diversity hotspots such as the insular Indo-Pacific and the coast of south-western Australia. The reason for this mismatch is that our analyses were focused on a specific set of seagrass species, for which sufficient occurrence data exist to make reliable predictions."

We have also improved and clarified the methodology throughout the manuscript.

The manuscript is focus from the beginning on the macrophyte-dominated habitats without framing the research in a more general biogeographic context of interest for a wide audience. In particular, the authors center the introduction on the mentioned ecosystems from the first sentence, without making any reference or comparison with other systems, marine or terrestrial, in regards of the biodiversity loss also experienced in those. I suggest start de introduction on the topic of global patterns of and losses of biodiversity worldwide and introduce the reader on the current knowledge on this topic. For this the authors may improve biological data by include the extensive records observed for macrophytes before 2015 (see comments below).

In relation to the above, I suggest fuse and shorten the first and second paragraphs of the introduction. I find well justified in the first paragraph the ecological relevance at a global scale of the studied systems, but do not see justification of a full paragraph on the contribution of macrophytes to carbon sequestration, a topic that is no mentioned afterwards. I suggest shortening drastically this paragraph.

RESPONSE #2: We have now framed the study into the broader biogeographic context by opening the Introduction with more general statements on projected patterns of biodiversity change in terrestrial and marine ecosystems (L34–37) (see Response #1). We have also fused and shortened paragraph 1 and 2 of the Introduction (L46–58).

Besides, the manuscript does not achieve general conclusions transferable to other ecological systems nor frame their findings on a biogeographic context in the discussion section. Please compare your results with those from Keith et al. (<https://doi.org/10.1111/geb.12132>), Tittensor et al. (<https://doi: 10.1038/nature09329>), or Short et al. (<https://doi.org/10.1016/j.jembe.2007.06.012>), among others. Given the extensive literature and discussion on biodiversity global patterns and trends (mostly terrestrial), I encourage the authors to include this and similar literature using a biogeographic approach. This may help in understanding how the predicted loss of these canopy-forming species would impact the global marine biodiversity.

RESPONSE #3: We have now expanded the comparison with previous studies on global patterns of diversity and distribution of marine macrophytes and other marine taxa by including the suggested references (L270–279):

“In general, present patterns of brown macroalgal species diversity are consistent with those previously described by others^{35,36}. Contrary to most marine coastal taxa⁵⁸, brown macroalgae show highest diversity in temperate regions such as the north-western Pacific, the north Atlantic and south-eastern Australia. Conversely, the highest species diversity of seagrasses is mainly found in the tropics^{37,59} (particularly in the Tropical Indo-Pacific bioregion). However, our map of current seagrass species diversity only partially overlaps with those of Short and colleagues^{37,59} given that our models did not identify prominent diversity hotspots such as the insular Indo-Pacific and the coast of south-western Australia. The reason for this mismatch is that our analyses were focused on a specific set of seagrass species, for which sufficient occurrence data exist to make reliable predictions.”

However, we have refrained from expanding the comparison to terrestrial systems because that milieu falls out of the scope of our manuscript.

The authors develop Species Distribution Modeling (SDMs) using Random Forest, a method linking presence records and pseudo-absences with the environmental layers of relevant environmental predictors. As mentioned in the introduction, this modeling is important for predicting habitat suitability. This modeling tool, largely use by biogeographers, has not been framed in this context and sound references of the theoretical base are not included. The term SDM, or equivalent, is not mentioned throughout the manuscript. Please give a brief explanation of what are these models and the state of the art, as there are not routinely used in marine systems.

RESPONSE #4: We now provide more information and clarify the modelling approach we used. Specifically, we now provide theoretical background on the random forest algorithm in the Results and Methods, as well as examples of its application in modelling species distributions in marine systems (e.g., L447–450):

“Random forest is a machine-learning method based on an ensemble of bootstrapped decision trees⁵². Due to the high accuracy of predictions and the ability to handle complex interactions and collinearity among predictors⁸¹, random forests have been increasingly used to model species distributions, including marine taxa^{82,83}.”

That said, we disagree that random forests would be something new or strange in the marine (or any other) context, or that they would be “*largely use[d] by biogeographers*”. Given the reviewer’s emphasis on general patterns and theory for different realms, we stress that random-forest methods are routinely used across all fields and realms of ecology — a statement we now support with additional references.

Regarding the timeframe used in this study, I do not see justification what records previous to 2015 have not been used, as most observation of macrophytes distribution comprised in GBIF, OBIS, and Assis` databases were taken before. The extensive literature on biogeography was reviewed by Lüning in its popular book of 1990, there is large information on the distribution of the studied species that has been omitted without justification why.

Besides, the data used in this study are affected by the heat wave of 2003 and later, as well as by the baseline warming observed before this date. Therefore, some historical populations may be missing because of the warming, and thus the realized niche captured in the SDMs may not represent the full conditions the species is able to cope with. Please justify the timeframe of this study and account that data are already affected by climate change.

RESPONSE #5: This is a misunderstanding of our approach. but we concede that this aspect was not clear in the previous version of the manuscript. We have now clarified that the dataset we used (Assis et al. 2020) includes records from 1665 to 2018. We only filtered the original version of the dataset to exclude species with few occurrence records (< 10), but we did use occurrences recorded before 2015 for our models.

As the authors are probably aware, there is uncertainty in the ways biodiversity may be predicted in the future. In particular stacking predicted maps of areas-of occupancy do not inform on potential biological interactions and dispersal restrictions impeding the species to survive. The debate on how to improve predictions of biodiversity is open, and thus I suggest the authors to mention this potential limitation and properly account for this topic in introduction, methods, and discussion. See for example Zhang et al. (<https://doi.org/10.1111/ddi.12970>).

RESPONSE #6: We agree that predicting biodiversity is challenging, and that stacking individual distribution maps has potential limitations, including not accounting for interspecific interactions. As we discuss in the text, the main goal of our approach is to explore changes in the potential future distribution and diversity of macrophytes; we are not aiming to predict future community composition. As we discuss in L350–351, not accounting for species interactions and potential dispersal restrictions makes our projections of future macrophyte ranges and diversity conservative. We have now emphasized these issues in the Discussion (L337–342):

“In addition, biotic interactions can modify brown macroalgal and seagrass assemblages through top-down mechanisms^{68,69}, and will therefore likely play a role in modulating the future distribution of both groups. For example, increased grazing pressure caused by the expansion of tropical herbivorous fishes has depleted brown algal cover in many temperate regions⁶⁸. Interspecific interactions between individual macrophyte taxa — such as competition or facilitation — might contribute additional complexity⁷⁰.”

In relation to the above, distributional contractions by climate change linked to stressful climatic and physical conditions, and thus predicted with relatively confidence. On the contrary, expansions to areas that are becoming climatically suitable are dependent on the ability to disperse and overcome the inhibiting biotic interactions. This matter is explained by biogeographers using the BAM diagram (https://www.youtube.com/playlist?list=PLJSfnPII0T5i_6Svb4L4VO-yAHwaQVFeU). Given that macroalgae are not good dispersers and seagrasses rely on clonal growth, I believe that macrophytes will find difficulties in expanding their limits. Post filtering using information on dispersal may improve the projections by SDMs (Anderson, [https://doi:10.1111/nyas.12264](https://doi.org/10.1111/nyas.12264)). Instead, the authors argue that the current prevalence (ratio of presences/absences) of the species may capture dispersal limitations. I do not see how prevalence

can inform on the elapsed time need for the species to reach the new area, that may be longer than the date of the projection. Post filtering the SDMs output using the dispersal rates of some model-species my help in identifying if the potential dispersal of relevant taxa may fell within the timeframe of the projections. I see clear theoretical differences between these two topics (dispersal Vs. prevalence-dominance) that are treated as a single response in paragraph starting in line 193, and in discussion in line 217. I believe prevalence is not a good proxy of dispersal, but please refer to previous references if so.

RESPONSE #7: We agree that post-filtering the species distribution model output with information on the species dispersal would be a powerful approach to model potential expansions into newly suitable areas. While we initially considered this approach, our decision to use prevalence as a proxy for future dispersal was dictated by the paucity of data on the dispersal ability and recruitment of most of the species we modelled — especially for brown macroalgae. Moreover, many other biotic and abiotic factors — including oceanic currents, wind speed, storms, temperature, nutrients, light and substrate availability, presence of competitors and grazers — can affect the dispersal and recruitment of both macrophyte groups. Given this complexity and the broad geographic and taxonomic coverage of our study, explicitly modelling macrophyte dispersal would be an endeavour too great and too speculative to be incorporated in our analyses.

We are aware of the limitations of using current prevalence rates as a proxy for the various potential factors affecting the ability to colonize suitable areas, as we outlined in the Discussion (L260–266). However, our ‘proportional expansion’ hypothesis provides an intermediate scenario between assuming no dispersal into novel suitable areas (hence, modelling macrophyte range erosion alone), and assuming full dispersal into novel suitable areas — two equally unrealistic scenarios.

Line 99- the authors are seeking for the “largest possible range where the species could occur” for calculating the extent of occurrence. For this I believe that data before 2015, particularly the extensive ones from the 70s, 80s and 90s, shall be considered. I see no justification for not using all the biological information available and thus suggest extending the database.

RESPONSE #8: As we explained in Response #5, the dataset we used (Assis et al. 2020) includes records from 1665 to 2018, and we have therefore included occurrences recorded before 2015 in ours analyses.

The lack of references in the “Results” and “Methods” sections is notorious. Line 98- lack of references on future climate scenarios, 100- lack of references on “extent of occurrence” and “area of occupancy” calculations, 104- lack of references of the validation method, 106- lack of references of the estimation of variance importance, 116- please add references to give support of the approach for predicting biodiversity. This is not solved in the “Methods” section where there are just two references on the niche modeling (numbers 28 and 76): 370- lack of references to random forest and SDMs, 388- lack of references of the protocol used, among others.

RESPONSE #9: We have now expanded the cited reference (especially in the Methods).

Line 110- How these results fit with the previous knowledge on the biodiversity global patterns of macrophytes and other relevant marine taxa?

RESPONSE #10: We now treat these results in more depth in the Discussion (L270–279):

“In general, present patterns of brown macroalgal species diversity are consistent with those previously described by others^{35,36}. Contrary to most marine coastal taxa⁵⁸, brown macroalgae show highest diversity in temperate regions such as the north-western Pacific, the north Atlantic and south-eastern Australia. Conversely, the highest species diversity of seagrasses is mainly found in the tropics^{37,59} (particularly in the Tropical Indo-Pacific bioregion). However, our map of current seagrass species diversity only partially overlaps with those of Short and colleagues^{37,59} given that our models did not identify prominent diversity hotspots such as the insular Indo-Pacific and the coast of south-western Australia. The reason for this mismatch is that our analyses were focused on a specific set of seagrass species, for which sufficient occurrence data exist to make reliable predictions.”

Line 120- The “gains” are probably better linked to the eltonian niche than to the one captured by SDMs. As argued above, gains cannot be given the same weight than losses, as are predicted with larger uncertainty. Please introduced this complexity in the ms. Are we really expecting biodiversity gains in the Artic and sub-Artic (coast of Iceland)? (Line 124). Besides, the species itself is not predicted to expand but its suitable habitat. Authors are projecting the species realized niche to the future rather than its presence. Please account for these limitations.

RESPONSE #11: Correct. We projected the geographic space where abiotic conditions will allow species to persist. The species true presence in this space will be further constrained by a suite of factors, including biotic conditions (e.g., species interactions) and by mobility. We concur that this makes range losses more certain than gains, because while the former can be determined by the loss of suitable abiotic conditions alone, the latter are also constrained by biotic and dispersal factors. We now emphasise these points in the Discussion (L291–294):

“However, diversity losses are predicted with greater certainty than gains, because while the former can be determined by the loss of suitable abiotic conditions alone, the latter are also constrained by biotic and dispersal components that we did not incorporate in our models.”

However, we obtained the species diversity maps by summing the area-of-occupancy maps produced by cropping the species-specific suitability maps with the extent of occurrence obtained from the occurrence records of each species based on an α -hull method (see Methods and Supplementary Fig. 1). In the area-of-occupancy maps, this restricts future suitable localities to the geographic area where the species can likely disperse (because it has already been recorded there), thereby partially buffering the issue related to dispersal. As for the biotic factors, we could not incorporate these in our models. Inclusion would introduce more uncertainty on the real future distribution of each macrophyte species, also affecting our projections of macrophyte diversity (explained in L345–349). However, the

influence of biotic interactions on species distributions might diminish at large geographical scales and coarse resolutions (Eltonian noise hypothesis; Soberón & Nakamura, 2009).

Fig 1. and similar- With such a low number of species of seagrasses (I understand is a small taxon in numbers), I do not see the need of use the logarithmic transformation of species richness, but the richness itself, as done in Short et al. (<https://doi.org/10.1016/j.jembe.2007.06.012>). Besides, results of this study contradict those from Short and other authors. For example, the diversity in the Mediterranean seems greater than that of the Tropical Indo-Pacific contradicting previous knowledge. Moreover, Fig. 1 show maxima logarithmic values in the Mediterranean, contradicting the text in line 114.

RESPONSE #12: We show seagrass species richness on a logarithmic scale for consistency with the species richness map of brown macroalgae. We now provide a map of seagrass species richness on a linear scale in the Supplementary Information (Supplementary Fig. 5, also shown below). We agree that our results on seagrass diversity are not fully in line with those of Short et al. (2007) and others. As we outline in the Discussion (L311–318), this is because we chose to focus our analyses on a specific set of seagrass species ($n = 22$ versus ~ 72 seagrass species globally described to date), for which sufficient occurrence data exist (≥ 10 occurrence records) to make reliable predictions. This is an important choice, because it allows us to focus on the more common and/or better-studied species for which current distributional data will likely be informative about underlying climatic drivers.

Supplementary Fig. 5 Present global macrophyte species diversity. We obtained present-day diversity by stacking individual area-of-occupancy maps (see Methods). **a** Shows brown macroalgae, **b** shows seagrasses. Unlike Fig. 1, the expected number of species in the figure is on a linear scale.

Line 140- please clarify and provide references supporting this approach of used pooled data en SDMs, were the raw data of all species pooled as the input of biological records in a Random Forest algorithm? or alternatively, did the authors combined the habitat suitability values of each species in a single final value?. I regret the exact methodology is not clear in this paragraph. In line 296 it said “combine”, please be precise.

RESPONSE #13: We now clarify our approach in the Methods (L450–457; see also Supplementary Fig. 1). We used the random forest algorithm for two reasons: (i) modelling species-specific habitat suitability and (ii) modelling global macrophyte habitat suitability. For *i*, we generated species-specific models based on all the occurrences of the target species. For *ii*, we generated a model to predict the overall suitability of a given locality to host brown algae (or seagrasses), regardless of species identity. In this second approach, we therefore used presence/absence of any brown algal (or seagrass) species as the dependent variable (i.e., we deemed a grid cell as a presence when it hosted at least one brown alga or seagrass record, regardless of species identity).

Line 259- authors point here the potential importance of considered other factors not included in this study. I do not understand why this is given more weight than to the discussion of the responses to the environmental predictors that were considered, an aspect that is not discussed in deep.

RESPONSE #14: We have now expanded the text describing the responses to the environmental predictors in the models (L321–325):

“Under all emissions scenarios, sea surface temperature and air temperature — together with surface incoming shortwave radiation — appeared to be the major determinant of macrophyte distribution. These results correspond with the general expectation that distributional shifts of marine taxa mainly track changes in ocean temperature¹.”.

Line 338- How you avoid autocorrelation between environmental predictors? this is well known to be an potential issue in SDMs and other multivariate approaches (Elith, <https://doi.org/10.1111/j.2041-210X.2010.00036.x>). The monthly means of the environmental parameters used in this study show necessary high autocorrelation. Please account for this important issue when modeling.

RESPONSE #15: The random forest algorithm is generally insensitive to collinearity and interactions among predictors (Breiman 2001; Cutler et al. 2007; Di Franco et al. 2016; Duffy et al. 2016). Unlike linear models that take into account all variables in relation to one another, decision trees in the random forests are constructed independently using a random subset of predictors, and thus tend to be de-correlated. Additionally, because the final outcome aggregates predictions from multiple trees, if some trees make predictions based on correlated variables, the final voting process tends to reduce the impact of individual trees potentially biased by variable correlation. However, to cope with any potential collinearity issues that might remain, we have now done a backward automatic variable selection for each model, reducing the number of predictors to 10.

Line 381- the models are calibrated for the recent-present day layers but cannot be trained using future conditions (SSP2-4,5, SSP3-7.0, AND SSP5-8.5) contrary to what is suggested in Methods. Training and predicting are mixed up and confused in the text: the models are first trained to the current-recent conditions and afterwards the final algorithm applied to future scenarios, please explain separately these two phases of the modeling process.

RESPONSE #16: We trained the models on the three different emissions scenarios for present-day layers for consistency (see also the method used by Strona and Bradshaw 2022).

Please end discussion making generalizations that arise necessary from comparisons with previous work. With the exception of the topic of global decline of macrophytes due to climate change, the authors do not include other general conclusion on how these results contribute to our understanding of the patterns and trends of global diversity. I strongly believe the manuscript has to give same weight to climate change science and biogeography.

RESPONSE #17: We have now added more general statements on how our results relate to patterns and trends in global biodiversity (L359-362):

“Overall, the substantial and geographically diverse redistribution of habitat-forming marine macrophytes projected in this study provides compelling evidence for the pervasive and intricate impacts of climate change on marine life^{1,74}.”

Reviewer 2

The authors use georeferenced species occurrence data for brown algae and seagrasses derived from online repositories as well as environmental data that are known to limit the distribution of these species. They then modelled how these environmental factors will change under future climate scenarios and what this would mean for the global distribution of these two groups of macrophytes. I am not a modeller so I can't comment on the validity of methodological approaches used, but the environmental data used to drive their models are appropriate and the outcomes of the analyses seem logical and align with similar regional studies. Overall this paper provides the first global analysis of how these two important groups of macrophytes are likely to shift their ranges into the future. As with any modelling approach there are a number of caveats that mean the predictions made in this manuscript may not come to pass. On the whole these are the same for any similar study and there is little that the authors could do, but in places on think the caveats need to be more strongly acknowledged. The authors also highlight that other studies separate model intertidal and subtidal species differently because drivers of their distribution are different. This study does not separate intertidal and subtidal species, but no justification is given for why this is appropriate in this study. This is a concern and I would argue that the authors should model these species separately as other studies have for the reasons stated or provide an evidence based justification for why it is okay in this instance. The number of species modelled (123) and the geographical coverage (12 eco-regions) make this study novel. I enjoyed reading the manuscript, but feel that authors need to explain how their approach of not separating intertidal and subtidal species is an appropriate approach given the different drivers of distribution for these species and therefore at this stage I believe that this manuscript can not be accepted for publication in its current state.

RESPONSE #18: We have now accounted for the difference between intertidal and subtidal species. We identified potential issues in making a pre-selection of subtidal *versus* intertidal species and in using different modelling approaches to project their future potential distribution at the global scale. In fact, some macrophytes (e.g., *Fucus vesiculosus*) occupy subtidal habitats in certain geographic areas and intertidal habitats in others. Instead, we expanded the model and let it identify potential differences in the drivers of distribution

between intertidal *versus* subtidal species. In fact, our previous model already included depth at high spatial resolution, so it was already accounting for potential differences in habitat preference among species to some extent. However, our new models now also include monthly surface air temperature — a variable that influences the distribution of intertidal marine macrophytes (Fragkopoulou et al. 2022). As we explained in Response #15, we also implemented an automated variable-selection procedure to minimize the ratio between number of independent variables and number of observations.

Lines 35 & 36 The references should come after the areal extent of the two habitats

RESPONSE #19: Done.

Line 54 It should be made clear that both DOC and POC can be transported to deep marine sediments

RESPONSE #20: Following the suggestion of Reviewer 1 (Response #2), we decided to shorten the paragraph, and we have removed reference to dissolved organic carbon and particulate organic carbon.

Line 64 Krumhansl et al 2016 PNAS 113: 13785-137990 should be cited at the end of this line and it might also be worth stating how many ecoregions have reported losses

RESPONSE #21: Done.

Lines 67-70 I suggest these sentences are restructured as the first suggests that there are no studies, when there are regional studies and then the second sentence acknowledges that this is the case.

RESPONSE #22: We have now rephrased the second sentence to improve clarity (L71-75):

“Currently available models projecting the future distribution of marine macrophytes apply to regional or local scales only, and/or to a limited set of species^{29–33} (but see a recent global study³⁴). Available studies focusing on the regional scale forecast substantial distributional shifts for both seagrasses^{30,32} and brown macroalgae³¹ by the end of this century.”

Line 78 It seems strange that only salinity and primary productivity are noted here. Light will be a key driver of these species distributions as will other interacting anthropogenic stressors. This is acknowledged in the discussion, but should be incorporated here as well.

RESPONSE #23: We now mention other drivers of marine macrophyte distribution (L90–94):

“Here, we hypothesize that future increases in sea surface temperature will force brown macroalgae and seagrasses to retreat from lower to higher (and cooler) latitudes, albeit with substantial regional variation modulated by differences in salinity and surface primary productivity^{42,43}, light availability,

water quality, and various other anthropogenic stressors acting at local and regional scales^{25,44}.”

Lines 81-82 Some macrophytes are likely to expand into the Arctic with reductions in sea-ice cover; but the picture is more complicated as outline in Filbee-Dexter et al Global and planetary Change 172: 1-14 review paper. I suggest caveating this sentence to acknowledge the more complex picture.

RESPONSE #24: We have now elaborated on the potential limiting factors to macrophyte expansion in the Arctic (L97–100):

“By contrast, the projected reduction in sea-ice cover and increasing sea temperatures might promote an expansion of macrophyte distribution into polar regions⁴⁶, although potentially constrained by the availability of suitable substrata, declines in salinity, and increases in turbidity expected from sea-ice melting⁴⁷.”

Line 123 There is also a hotspot of loss on the west coast of Australia that deserves a mention, particularly as this area supports a high diversity of seagrass species.

RESPONSE #25: We now mention the hotspot of loss in Western Australia in the Results (L151).

Lines 129-131 I suggest stating what the losses under the other climate change scenarios are and refer the reader to the SOM

RESPONSE #26: Done (L162–163):

“These figures reach $6.5\% \pm 0.2\%$ and $6.7\% \pm 0.5\%$, respectively, under SSP5-8.5 (Supplementary Fig. 7a, b).”

Line 133 Add negatively before impacted

RESPONSE #27: In light of the results obtained with the new, improved models, we have removed the sentence because it was no longer relevant.

Line 134 I suggest adding that these changes are particularly stark from 20 degrees north

Line 135-137 I suggest it might be worth highlighting the gains and then losses around 20 degrees south

RESPONSE #28: We have now described them more extensively (L164–172):

“When considering latitudinal patterns of diversity change throughout this century (Fig. 2c,d; Supplementary Fig. 6c,d; Supplementary Fig. 7c,d), the loss of brown macroalgae diversity appears particularly stark beyond 40° N and in the entire Southern Hemisphere, while gains mainly occur between 0–20° N (scenario SSP3-7.0; Fig. 2c,d). The loss of seagrass diversity appears more severe between 25–40° in

both hemispheres, while gains occur in the tropics (around 20° N and 20° S) and beyond 50° N in the Northern Hemisphere (scenario SSP3-7.0; Fig. 2c, d).

Conversely, both groups are projected to face widespread diversity loss in the tropics under a more pessimistic emissions scenario (SSP5-8.5; Supplementary Fig. 7c,d).”

Lines 164-166 It isn't clear to whether the values in parentheses are the increases/ decreases between 2015-2100 or the range of values the models provided. Please clarify

RESPONSE #29: For each marine region, the two values in parentheses represent the share (%) of global macrophyte habitat in the region (i) in 2015 and (ii) in 2100. We have now clarified this.

Lines 241-243 From my understanding there is increasing concern of brown macroalgae reaching the Antarctic via the rafting of, particularly species with air bladders, that have been dislodged bearing reproductive material. It may be worth acknowledging this as a method of dispersal, but not one that would be picked up with this modelling approach.

RESPONSE #30: We have modified the sentence accordingly (L304–307).

“Conversely, our models did not predict a strong expansion of macrophyte habitat in the Southern Ocean, which is consistent with the independent expectation arising from the biogeographic isolation of the Antarctic continent⁶², although dispersal to this region might occur via rafting for some species⁶³. ”

Line 245 It is only under the worst case scenario that any brown algal species losses most of their range. This should be acknowledged.

RESPONSE #31: We have removed the sentence from the Discussion.

*Line 249 I believe that *Macrocystis pyrifera* has a listing designation in Australia that might be worth acknowledging if this species is included in the analysis.*

RESPONSE #32: In light of the results obtained with the new, improved models, we have removed the paragraph because it was no longer relevant.

Figure 2 It should be stated in the figure legend what the blue and red represent in terms of gains and losses

RESPONSE #33: Done.

Figure 4 If this was upped to a higher ability how many regions would be excluded. I would be tempted to use as a high a probability as possible e.g. $p > 0.9$ as described in the text around Fig 3 and only use this lower value for the regions with limited suitable cells unless this is the case across all/majority of regions and if so this needs to be state.

RESPONSE #34: We now provide a separate figure in the Supplementary Information (Supplementary Fig. 8, also shown below) where we display trajectories of highly suitable habitat only ($p > 0.9$), excluding a total of three marine regions. However, we have refrained from using two different thresholds within the same figure in the main text, because it would be difficult to interpret the bar plots in panel b.

Supplementary Fig. 8 Variation in the extent of suitable macrophyte habitat ($p > 0.9$) across marine regions (emissions scenario SSP3-7.0). We calculated global macrophyte habitat suitability using a machine-learning approach (see Methods) for the period 2015–2100, and applied a threshold of $p = 0.9$ to select only highly suitable macrophyte habitat (only marine regions where highly suitable macrophyte habitat is present are shown). **a** Variation in macrophyte habitat extent (km^2) for brown macroalgae (purple) and seagrasses (pink) within each marine region, aggregated every 5 years. **b** Comparison of the percentage of global suitable macrophyte habitat in each marine region between 2015 and 2100 for brown macroalgae (upper bar plots) and seagrasses (lower bar plots). Colours refer to marine regions as shown in **a**. Square brackets show total global suitable habitat extension.

SOM Fig 3 and 4 are only discussed in the discussion and all SOM figures should be numbered in the order in which they appear.

RESPONSE #35: We have added references to all the Supplementary figures in the Results and numbered them in the order in which they appear.

Reviewer 3

This study uses habitat suitability modelling and climate projections to show how global distributions of large brown macroalgae (LBMs) and seagrasses are likely to change over the next 70 years or more. The approach is solid, if a little opaque in parts, with the findings only completely clear after a close study of the methods. Species with sufficient records to establish credible global distributions are modelled individually and the results combined to show how species diversity and total suitable habitats might change. The second part of the work models all seagrasses and LBMs together to produce similar projections to those from the combined individual species models, without adding much to the work. Dropping the total macrophyte modelling would allow for greater focus on how the findings are supported by the methodology. The changes in macrophyte results shown in Figs 3 to 5 could be produced from the summation of the species-specific models.

Some confusion for this reader was generated by the very similar-sounding terms for quite different measures of species ranges and suitable habitats. 'Extent of occurrence' covers the range of observations of each species, while 'area of occupancy' refers the overlap of 'extent of occurrence' and modelled maps of present-day and predicted future suitable habitat (lines 389-393). Future 'area of occupancy' can never therefore exceed present-day 'extent of occurrence'. 'Area' is also used interchangeably to mean the summed extent of a species distribution and the distribution itself.

RESPONSE #36: We have now clarified the different purposes and interpretations of the two approaches, paying particular attention to the potential confounding issues associated with our chosen terminology. Regardless, we have thoroughly explored the reviewer's suggestions, and summed species-specific area-of-occupancy maps to obtain global habitat suitability maps, both considering the mean and maximum habitat suitability (p) value per grid cell (among the suitability values of all species present in each cell). Confirming our statement above, the resulting patterns of habitat suitability are substantially different from those that we obtained from the generic habitat suitability model. We show the maps for the year 2015 (scenario SSP3-7.0) for comparison below.

We are aware that modelling generic macrophyte habitat suitability separately might seem redundant. However, the two analyses convey substantially different messages. This is because the generic habitat model, by being trained on the full set of conditions, does not necessarily represent the 'sum' of the individual models; instead, it identifies an independent set of non-linear relationships linking environmental variables to the occurrence of generic macrophyte habitat. This model is also expected to be statistically more robust than the individual models, because of the much larger pool of training observations.

Global macrophyte habitat suitability maps obtained with different approaches (year 2015, scenario SSP3-7.0). **a–b** maps obtained from the global habitat suitability model we used; **c–d** maps obtained from the summation of species-specific area-of-occupancy maps, considering the mean suitability value (p) per cell; **e–f** maps obtained from the sum of species-specific area-of-occupancy maps, considering the maximum suitability value (p) per cell. **a,c,e** brown macroalgae ($n = 185$ species); **b,d,f** seagrasses ($n = 22$ species). Coloured scalebars show habitat suitability (p).

Models of future distributions assumed no expansion of ranges (line 388). Adopting this decision meant that the focus was only the loss of present-day ranges, with largely negative impacts on future diversity. For these largely temperate species with no tropical counterparts to replace them with warming, declining diversity would be expected in their present-day temperate diversity hotspots. But the diversity change calculated here (lines 395-396) does not take account of potential gains from range expansions of warm-affinity species within temperate zones, inflating the likely losses in diversity.

‘Proportional expansion’ (lines 411), the assumption that species will continue to occupy a constant proportion of the predicted globally suitable habitat, is a fair enough proposition and better than that newly suitable habitat will be instantly occupied once it becomes available through climate change.

RESPONSE #37: We agree that limiting range expansions to the extent of occurrence of each species might inflate diversity losses, giving a more pessimistic scenario of diversity change. However, we also explored potential expansions beyond the limits of each species’ extent of occurrence under the ‘proportional expansion’ hypothesis (Fig. 5).

lines 26-27. Losses. It is not yet clear at this point that gains in range and suitable habitats are excluded, and expansions focussed on macrophyte habitat in general.

RESPONSE #38: We have now reworded the sentence to clarify that we are referring to different aspects of the future of macrophytes — specifically, their local diversity/species richness, loss of current range (which, by definition excludes future gains in areas outside it), and global loss of habitat (L27–32):

“... we estimate that by 2100, local macrophyte diversity will decline by 3–4% on average, with 17 to 22% of localities losing at least 10% of their macrophyte species globally. The current range of macrophytes will be eroded by 5–6%, and highly suitable macrophyte habitat will be substantially reduced globally (78–96%). Global macrophyte habitat will shift among marine regions, with a high potential for expansion in polar regions.”

lines 34-75. This is a nice summary of the literature on the main drivers of change in these groups, but the number of references could be reduced.

RESPONSE #39: We have shortened the Introduction (see Response #2) and the number of references.

line 120. The odd gains of species diversity for LBM in the northwest Indian Ocean are hard to understand (Fig. 2b). Are existing tropical species predicted to expand to areas previously unoccupied? Similarly, gains for seagrasses east Africa (Fig. 2d) need some explanation.

RESPONSE #40: The diversity in neighbouring areas appears to be stable. An increase in diversity in the north-western Indian Ocean (blue cells) might be due to an increase in suitable habitat for local species (i.e., within their extent of occurrence). As for seagrasses, our new and improved models did not show any gains in diversity in East Africa.

Figure 3. The greater losses of seagrass habitat extent under the less-extreme climate scenarios of SSP2 and SSP3 than SSP5 are difficult to reconcile. Why is that happening?

RESPONSE #41: The results we obtained with the new models do not show such patterns (Fig. 3).

Figure 4. Information presented here is essentially similar to Figs 1 and 2, with declines in tropical regions, stasis in cold temperate regions and increases in polar regions where suitable habitats may exist – here the Arctic only.

RESPONSE #42: As we stated in Response #36, modelling global macrophyte habitat suitability separately conveys a different kind of information than the sum of single area-of-occupancy maps. The magnitude of regional shifts in the extent of suitable habitat for macrophytes (Fig. 4) does not necessarily mirror changes in macrophyte species diversity in that marine region (Fig. 1). For example, in the tropical Atlantic (Fig. 4a), suitable habitat for brown macroalgae ($p > 0.6$) is projected to diminish by the end of the current century, while this negative trend is not evident in the map of brown macroalgal species diversity (Fig. 1b), where both diversity gains and losses occur in the region. Likewise, the western Indo-Pacific is projected to face a high loss of suitable habitat for seagrasses (Fig. 4a), which does not show in the map of seagrass species diversity (Fig. 1d).

Figure 5. As before, distinction between contraction of existing distributions (solid lines give changes in area of occupancy) and potential expansion into newly suitable areas (dashed lines) makes it difficult to see a net gain or loss in LDMs and seagrasses.

RESPONSE #43: We have now clarified the figure legend:

“Solid lines represent mean variation in macrophyte current area of occupancy (assuming no expansion beyond the species’ extent of occurrence); dashed lines represent mean expected change in macrophyte area of occupancy assuming the current ratio between occupied and climatically suitable range will remain constant (*proportional expansion*; see Methods); this allows for expansion beyond the limits of a species’ extent of occurrence.”

References cited in this response

- Assis, J., Fragkopoulou, E., Frade, D., Neiva, J., Oliveira, A., Abecasis, D. *et al.* (2020). A fine-tuned global distribution dataset of marine forests. *Scientific Data*, 7, 119. doi:10.1038/s41597-020-0459-x
- Breiman, L. (2001). Random forests. *Machine learning* 45, 5–32
- Cutler, D.R., Edwards Jr, T.C., Beard, K.H., Cutler, A., Hess, K.T., Gibson, J. *et al.* (2007). Random forests for classification in ecology. *Ecology*, 88, 2783-2792. doi:10.1890/07-0539.1
- Di Franco, A., Thiriet, P., Di Carlo, G., Dimitriadis, C., Francour, P., Gutiérrez, N.L. *et al.* (2016). Five key attributes can increase marine protected areas performance for small-scale fisheries management. *Scientific Reports*, 6, 38135. doi:10.1038/srep38135
- Duffy, J.E., Lefcheck, J.S., Stuart-Smith, R.D., Navarrete, S.A. & Edgar, G.J. (2016). Biodiversity enhances reef fish biomass and resistance to climate change. *Proceedings of the National Academy of Sciences of the USA*, 113, 6230-6235. doi:10.1073/pnas.1524465113
- Fragkopoulou, E., Serrão, E.A., De Clerck, O., Costello, M.J., Araújo, M.B., Duarte, C.M. *et al.* (2022). Global biodiversity patterns of marine forests of brown macroalgae. *Global Ecology and Biogeography*, 31, 636-648. doi:10.1111/geb.13450
- Short, F., Carruthers, T., Dennison, W. & Waycott, M. (2007). Global seagrass distribution and diversity: A bioregional model. *Journal of Experimental Marine Biology and Ecology* 350, 3-20
- Soberón, J., & Nakamura, M. (2009). Niches and distributional areas: concepts, methods, and assumptions. *Proceedings of the National Academy of Sciences*, 106(supplement_2), 19644-19650.

Strona, G. & Bradshaw, C.J.A. (2022). Coextinctions dominate future vertebrate losses from climate and land use change. *Science Advances*, 8, eabn4345. doi:10.1126/sciadv.abn4345

REVIEWERS' COMMENTS

Reviewer #1 (Remarks to the Author):

I have reviewed my previous comments to the original submission, and the new reviewed manuscript in comparison to the previous version, and I believe the authors have done major changes following the suggestions raised in the review. The manuscript has gain interest for a wide audience and improved in clarity. Besides, my concerns have been answered whit sound arguments, and the information requested is now included in the revised version, as for example the clarification on the time frame of the presence records, or the previous knowledge on the latitudinal patterns of macrophytes, among other major changes. I am thus changing my suggestion to “publishable” after few additional issues that I listed below.

The new paragraph in the introduction dealing with the biogeographic patterns of the macrophytes need to be "polished" to explain the ideas with precision:

- line 80. “Distinct” in what manner? these two groups exhibit contrasting latitudinal patterns in biodiversity, please explain in brief, the idea is left "open".
- line 82, what complex processes do the authors refer here?, please explain.
- line 83, what environmental requirements? please clarify. This contrast with the idea that both groups dependent on light and temperature in line 83, and therefore, depending on similar environmental requirements?.

I believe the set of environmental predictors probably show autocorrelation, as in particular sea surface and air temperatures, thought the authors do not include estimations on this autocorrelation (as for example Pearson Coefficients between pairs or Variation Inflation Factors VIFs). Nevertheless, the authors provide references supporting Random Forest can perform in such conditions in their rebuttal letter. There are indeed some applications/packages dealing with this potential issue, as for example:

<https://blasbenito.github.io/spatialRF/#reducing-multicollinearity-in-the-predictors>. I thus believe the algorithm and statistical approach is correct but would suggest the authors to include such information on the magnitude of autocorrelation, and information on the precise software or package used to perform the RF analysis, the options of parametrization used to lover this potential issue, and the technical details related to this concern.

Reviewer #2 (Remarks to the Author):

The authors have done an excellent job of incorporating the three reviewers comments and this has significantly improved the manuscript. Where the comments of reviewers have not been incorporated this is either because that material is no longer in the manuscript or the authors, in my opinion, provide a strong justification for not doing so. This manuscript is a very interesting read and is novel in providing the first global assessment of likely changes in distribution and richness of key marine ecosystem engineers. The authors have taken the time in the current version to more fully acknowledge the limitations of the techniques used, but this isn't a failing of the paper just the natural limitations of SDM modelling - an approach widely used for predicting for species distributions. I believe this manuscript will be of interest to a broad audience and I recommend publication with some very minor corrections (see below). I would also like to thank the authors for their very clear response to reviewers. It made it very easy to link the reviewer comment with the response, especially with the inclusion of the change in text.

Minor comments

Line 29 I suggest removing globally as somewhat redundant

Line 123 I would indicate sea and air temperature

Line 127 I would use light instead of shortwave radiation as it is more accessible

Line 271 Suggest deleting 'over the global range' and replace with 'across ecoregions' as it is more reflective of the comparisons undertaken

Reviewer #3 (Remarks to the Author):

My points concerned the need for clearer terminology and a more upfront message that the study emphasised loss of existing distributions in the way that the models were set up (Response #37). The reasons for the latter focus on losses are clear enough, but this aspect is lost in the detail – not obvious from the Abstract for example. It was good to see the exploration of the differences between the maps of summed suitability models and that of overall macrophyte suitability. The figure on p15 of the rebuttal would be a useful addition to the supplement (after Supplementary Fig. 3). The difference between the two approaches (stacked SDMs and modelled macrophyte suitability) highlights an important

issue: data-poor species models may be more likely to predict presence in unsuitable areas and less likely in more suitable areas than the data-rich all macrophyte one. Stacked species models may be more likely to give higher species diversity (Fig. 1a) in unsuitable areas as a consequence. A final point is that the new Fig. 3 still shows greater losses of seagrasses for less extreme climate scenarios SSP2 and SSP3 than SSP5, especially for higher suitability thresholds (0.7-0.9). There might be a reason for this but a comment should be made.

Reviewer 1

I have reviewed my previous comments to the original submission, and the new reviewed manuscript in comparison to the previous version, and I believe the authors have done major changes following the suggestions raised in the review. The manuscript has gain interest for a wide audience and improved in clarity. Besides, my concerns have been answered whit sound arguments, and the information requested is now included in the revised version, as for example the clarification on the time frame of the presence records, or the previous knowledge on the latitudinal patterns of macrophytes, among other major changes. I am thus changing my suggestion to “publishable” after few additional issues that I listed below.

The new paragraph in the introduction dealing with the biogeographic patterns of the macrophytes need to be "polished" to explain the ideas with precision:

-line 80. “Distinct” in what manner? these two groups exhibit contrasting latitudinal patterns in biodiversity, please explain in brief, the idea is left "open".

RESPONSE #1: We have now clarified the sentence (L81–83):

“Brown macroalgae and seagrasses display distinct global distributions, both in terms of latitudinal patterns of biodiversity and spatial location of diversity hotspots^{35–37}.”

-line 82, what complex processes do the authors refer here?, please explain.

RESPONSE #2: The biogeographic processes that drove the current distribution of brown macroalgae and seagrasses are explained in detail in the cited literature (Bolton 2010 and Larkum et al. 2018). We chose not to delve into the topic extensively in the text, but we have now added a short sentence to clarify to which general processes we refer (L83–85):

“Such differences arise both from long and complex biogeographic processes — such as distinct patterns of speciation and dispersal^{38,39}, and from taxon-specific ecological and environmental requirements.”

-line 83, what environmental requirements? please clarify. This contrast with the idea that both groups dependent on light and temperature in line 83, and therefore, depending on similar environmental requirements?.

RESPONSE #3: Temperature and light are primary drivers of the distribution of both groups. However, in general, brown macroalgae and seagrasses have different optima and tolerance limits both for temperature and light. Most species of brown macroalgae are in temperate waters (Fragkopoulou et al. 2022), while the highest diversity of seagrasses occurs in warm, tropical waters (Short et al. 2007). Moreover, seagrasses generally have higher minimum light requirements than brown macroalgae (Duarte 1991).

I believe the set of environmental predictors probably show autocorrelation, as in particular sea surface and air temperatures, thought the authors do not include estimations on this autocorrelation (as for example Pearson Coefficients between pairs or Variation Inflation Factors VIFs). Nevertheless, the authors provide references supporting Random Forest can perform in such

conditions in their rebuttal letter. There are indeed some applications/packages dealing with this potential issue, as for example: <https://blasbenito.github.io/spatialRF/#reducing-multicollinearity-in-the-predictors>. I thus believe the algorithm and statistical approach is correct but would suggest the authors to include such information on the magnitude of autocorrelation, and information on the precise software or package used to perform the RF analysis, the options of parametrization used to lower this potential issue, and the technical details related to this concern.

RESPONSE #4: We have further addressed the issue of autocorrelation among variables as requested. We have calculated the average R^2 between each pair of variables used in the single-species models before and after variable selection for each emissions scenario (SSP2-4.5, SSP3-7.0, SSP5-8.5). We report in the boxplot below the distribution of the mean R^2 for the pairwise variable comparison for each model in the three emissions scenarios. On average, pairwise variable correlation was < 0.3 . The variable selection procedure reduced the number of variables per model by 56% on average and removed the most correlated variables in 82% of models. However, as we discussed in our previous response to reviewers, in principle, autocorrelation among predictors does not affect the performance of the random forest algorithm (Breiman 2001; Dormann et al. 2013). We now provide further details on the variable selection procedure in the Methods (L473–481):

“In the individual models, we applied a variable-selection procedure where we iteratively removed the 10 least-important variables (starting from the full model), recomputing for each model the accuracy as an out-of-bag validation score⁵² and variable importance as the mean accumulation of impurity decrease. We then selected the most accurate model. The procedure substantially reduced the number of predictors in each model ($55.8\% \pm 28.5$ on average). This also resulted in the exclusion of the most correlated independent variables in 82% of the models.”

Autocorrelation (R^2) among variables selected in the species-specific models.

Distribution of the mean R^2 calculated among each pair of variables used in every species-specific model after variable selection (for the year 2015) for all the three emissions scenarios (SSP2-4.5, SSP3-7.0, SSP5-8.5).

Reviewer 2

The authors have done an excellent job of incorporating the three reviewers comments and this has significantly improved the manuscript. Where the comments of reviewers have not been incorporated this is either because that material is no longer in the manuscript or the authors, in my opinion, provide a strong justification for not doing so. This manuscript is a very interesting read and is novel in providing the first global assessment of likely changes in distribution and richness of key marine ecosystem engineers. The authors have taken the time in the current version to more fully acknowledge the limitations of the techniques used, but this isn't a failing of the paper just the natural limitations of SDM modelling - an approach widely used for predicting for species distributions. I believe this manuscript will be of interest to a broad audience and I recommend publication with some very minor corrections (see below). I would also like to thanks the authors for their very clear response to reviewers. It made it very easy to link the reviewer comment with the response, especially with the inclusion of the change in text.

Minor comments

Line 29 I suggest removing globally as somewhat redundant

RESPONSE #5: Removed.

Line 123 I would indicate sea and air temperature

RESPONSE #6: Done.

Line 127 I would use light instead of shortwave radiation as it is more accessible

RESPONSE #7: Done.

Line 271 Suggest deleting 'over the global range' and replace with 'across ecoregions' as it is more reflective of the comparisons undertaken

RESPONSE #8: Replaced.

Reviewer 3

My points concerned the need for clearer terminology and a more upfront message that the study emphasised loss of existing distributions in the way that the models were set up (Response #37). The reasons for the latter focus on losses are clear enough, but this aspect is lost in the detail – not obvious from the Abstract for example.

RESPONSE #9: We had already discussed this aspect in the Results (L233–236):

“Changes in the extension of the area of occupancy of each species can only be due to variation in habitat suitability within the boundaries of its present-day extent of

occurrence (hence, we assume no dispersal in areas where the species has not yet been recorded).”

However, we have now highlighted the issue also in the paragraph on present and future macrophyte species diversity (L136–139):

“We obtained area-of-occupancy maps by determining habitat suitability within the boundaries of each species’ present-day extent of occurrence (assuming no dispersal in areas where the species has never been recorded).”

It was good to see the exploration of the differences between the maps of summed suitability models and that of overall macrophyte suitability. The figure on p15 of the rebuttal would be a useful addition to the supplement (after Supplementary Fig. 3).

RESPONSE #10: We have now added the figure to the Supplementary Information (Supplementary Fig. 8).

The difference between the two approaches (stacked SDMs and modelled macrophyte suitability) highlights an important issue: data-poor species models may be more likely to predict presence in unsuitable areas and less likely in more suitable areas than the data-rich all macrophyte one. Stacked species models may be more likely to give higher species diversity (Fig. 1a) in unsuitable areas as a consequence.

RESPONSE #11: We agree that models fitted with few occurrences might not yield accurate predictions compared to more data-rich models. However, in general, our species-specific models had high accuracy, with an average out-of-bag validation score of 0.987 ± 0.001 (\pm standard error; minimum = 0.773).

A final point is that the new Fig. 3 still shows greater losses of seagrasses for less extreme climate scenarios SSP2 and SSP3 than SSP5, especially for higher suitability thresholds (0.7-0.9). There might be a reason for this but a comment should be made.

RESPONSE #12: We have now added a comment in the Results (L197–200):

“Under the most pessimistic emissions scenario (SSP5-8.5, Fig.3), the decline in highly suitable macrophyte habitat is projected to be more severe for brown macroalgae (81%), but not for seagrasses (92.1%).”

References cited in this response

- Bolton, J. J. The biogeography of kelps (Laminariales, Phaeophyceae): a global analysis with new insights from recent advances in molecular phylogenetics. *Helgol. Mar. Res.* 64, 263–279 (2010)
- Breiman L (2001) Random forests. *Mach Learn* 45(1):5-32. doi:10.1023/A:1010933404324
- Dormann CF, et al. (2013) Collinearity: a review of methods to deal with it and a simulation study evaluating their performance. *Ecography* 36(1):27-46. doi:10.1111/j.1600-0587.2012.07348.x
- Duarte, C. M. (1991). Seagrass depth limits. *Aquatic botany*, 40(4), 363-377.

- Fragkopoulou, E., Serrão, E. A., De Clerck, O., Costello, M. J., Araújo, M. B., Duarte, C. M., ... & Assis, J. (2022). Global biodiversity patterns of marine forests of brown macroalgae. *Global Ecology and Biogeography*, 31(4), 636-648.
- Larkum, A. W. D., Waycott, M. & Conran, J. G. Evolution and Biogeography of Seagrasses. in *Seagrasses of Australia: Structure, Ecology and Conservation* (eds. Larkum, A. W. D., Kendrick, G. A. & Ralph, P. J.) 3–29 (Springer International Publishing, 2018). doi:10.1007/978-3-319-71354-0_1.
- Short, F., Carruthers, T., Dennison, W., & Waycott, M. (2007). Global seagrass distribution and diversity: a bioregional model. *Journal of experimental marine biology and ecology*, 350(1-2), 3-20.